# Gut microbiota mediate the FGF21 adaptive stress response to chronic dietary protein-restriction in mice

Anthony Martin[1], Gertrude Ecklu-Mensah [2], Connie W. Y. Ha[1], Gustaf Hendrick[1], Donald K. Layman [3], Jack Gilbert[2] & Suzanne Devkota [1✉]

Chronic dietary protein-restriction can create essential amino acid deficiencies and induce metabolic adaptation through the hepatic FGF21 pathway which serves to maintain host fitness during prolonged states of nutritional imbalance. Similarly, the gut microbiome undergoes metabolic adaptations when dietary nutrients are added or withdrawn. Here we confirm previous reports that dietary protein-restriction triggers the hepatic FGF21 adaptive metabolic pathway and further demonstrate that this response is mediated by the gut microbiome and can be tuned through dietary supplementation of fibers that alter the gut microbiome. In the absence of a gut microbiome, we discover that FGF21 is de-sensitized to the effect of protein-restriction. These data suggest that host-intrinsic adaptive pathways to chronic dietary protein-restriction, such as the hepatic FGF21 pathway, may in-fact be responding first to adaptive metabolic changes in the gut microbiome.

[1] F. Widjaja Foundation Inflammatory Bowel & Immunobiology Research Institute, Cedars-Sinai Medical Center, Los Angeles, CA, USA. [2] Department of Pediatrics and Scripps Institution of Oceanography, University of California San Diego, La Jolla, CA, USA. [3] Department of Food Science and Human Nutrition, University of Illinois, Urbana, IL, USA. ✉email: Suzanne.Devkota@cshs.org

Global dietary behavior and nutrition status varies across populations based on geography, socioeconomics, and religious practices. For example, non-Western populations trend toward consumption of low-protein, high-fiber diets, and in extreme cases, abject protein deficiency in infants results in kwashiorkor[1–4]. Furthermore, modern dietary practices in Western societies may also inadvertently induce protein deficiencies particularly when strictly plant-based diets are consumed[5]. While plant proteins as a group possess the full repertoire of amino acids (AA), individually, most plant proteins are limiting or deficient in at least one AA[6]. Therefore, chronically inadequate consumption of diverse plant protein sources on a strictly plant-based diet may unintentionally increase an individual's risk for essential AA deficiencies. In turn, strategies to mitigate the adverse physiological effects of prolonged protein deficiency are needed as interventions aimed at simply increasing protein intake are not always a tenable option due to cost or availability.

Metabolic strategies to preserve fitness in times of nutrient deficiency exist across all species. In the context of dietary protein -restriction (PR), an adaptive metabolic stress response driven by the hepatic endocrine molecule, FGF21, promotes increased food intake[7] and energy expenditure while reducing body weight and adiposity[8–13]. Elevated FGF21 in times of protein scarcity is therefore necessary and appropriate. There appears to be a specificity of FGF21 for sensing dietary protein status as several studies have demonstrated FGF21 is elevated when total protein or individual AAs are restricted, even if calories are not. Other dietary scenarios such as ketogenic diets and starvation can also increase FGF21 levels, however levels increase further when the ratio of protein to the other macronutrients in these diets is further reduced[14,15]. The therapeutic use of FGF21 analogues has garnered intense interest in the context of treating metabolic disorders due to its demonstrated effects on decreasing plasma glucose and triglycerides while preventing weight gain in mice fed a high-fat diet[16,17]. Therefore, the physiological impacts of FGF21 and its overall systemic importance are numerous in part due to its expression in several tissues including pancreas, adipose, and liver[18]. However, circulating levels of FGF21 are largely attributed to hepatic production.

Several studies have reported the increase of plasma FGF21 levels in response to diminished dietary AA intake[11,19–23] and it thus serves as a useful biomarker for determining host protein status during suspected protein deficiency. One aspect that has been missing from these discussions is the role of the gut microbiome on host nutrient sensing, given that the gut microbiome is likely undergoing adaptation concurrently[24,25]. Chronic intake of nutritionally imbalanced diets can also alter host-microbe interactions by creating a selection pressure on gut microbial communities that skews toward the most adaptable organisms. This has been demonstrated in a multitude of diet studies, many of which proving that microbial metabolites produced in response to host diet have a significant impact on host physiology[26–29]. Given that the gut microbiota are nutrient sensors in their own right, it is important to understand whether the gut microbiota may be integrated into the FGF21 response and thus may be another point of manipulation for therapeutic benefit in times of PR.

In the following study, we demonstrate that the FGF21 response to PR is abrogated in germ-free (GF) mice. Furthermore, we find PR diets supplemented with fermentable or non-fermentable dietary fiber can differentially suppress the FGF21 stress response in conventional mice by promoting the growth of certain bacterial taxa with concomitant effects on weight gain and downstream markers of protein status. These data suggest that FGF21 may ultimately be responsive to adaptive changes in the gut microbiota due to PR.

## Results

**Cellulose and inulin differentially mitigate the FGF21 metabolic stress response elicited by dietary PR.** Mice were provided one of eight diets, varying in protein amount and fiber composition, for 21 days to determine the effect of PR on the FGF21 pathway while investigating the efficacy of dietary interventions to mitigate the stress response (Fig. 1). Consistent with previous reports, we demonstrate that circulating levels and hepatic gene expression of FGF21 were elevated in mice fed a protein-restricted diet (PR, 10% protein by weight) relative to mice fed a control, protein-sufficient diet (PS, 18% protein by weight) (Fig. 2a, b; black bars). We sought, therefore, to find dietary

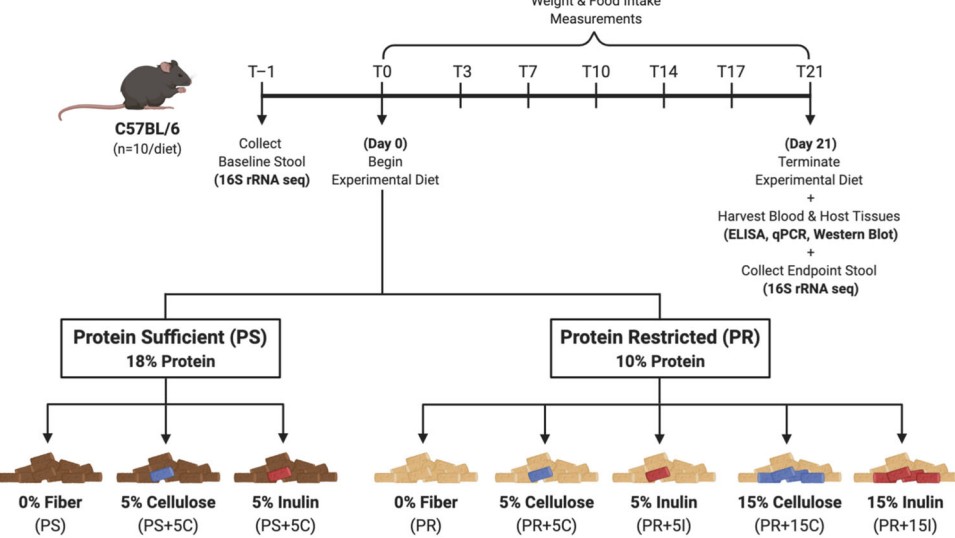

**Fig. 1 Schematic of study design.** Mice were fed either a protein-sufficient (PS) or protein-restricted (PR) diet for 21 days. PS diets were supplemented with 5% cellulose (PS + 5C) or 5% inulin (PS + 5I). PR diets were supplemented with either 5% cellulose or 15% cellulose (PR + 5C or PR + 15C, respectively) or 5% inulin or 15% inulin (PR + 5I or PR + 15I, respectively) All diets are isocaloric. Each mouse represents an independent sample replicate. This illustration was generated in BioRender.

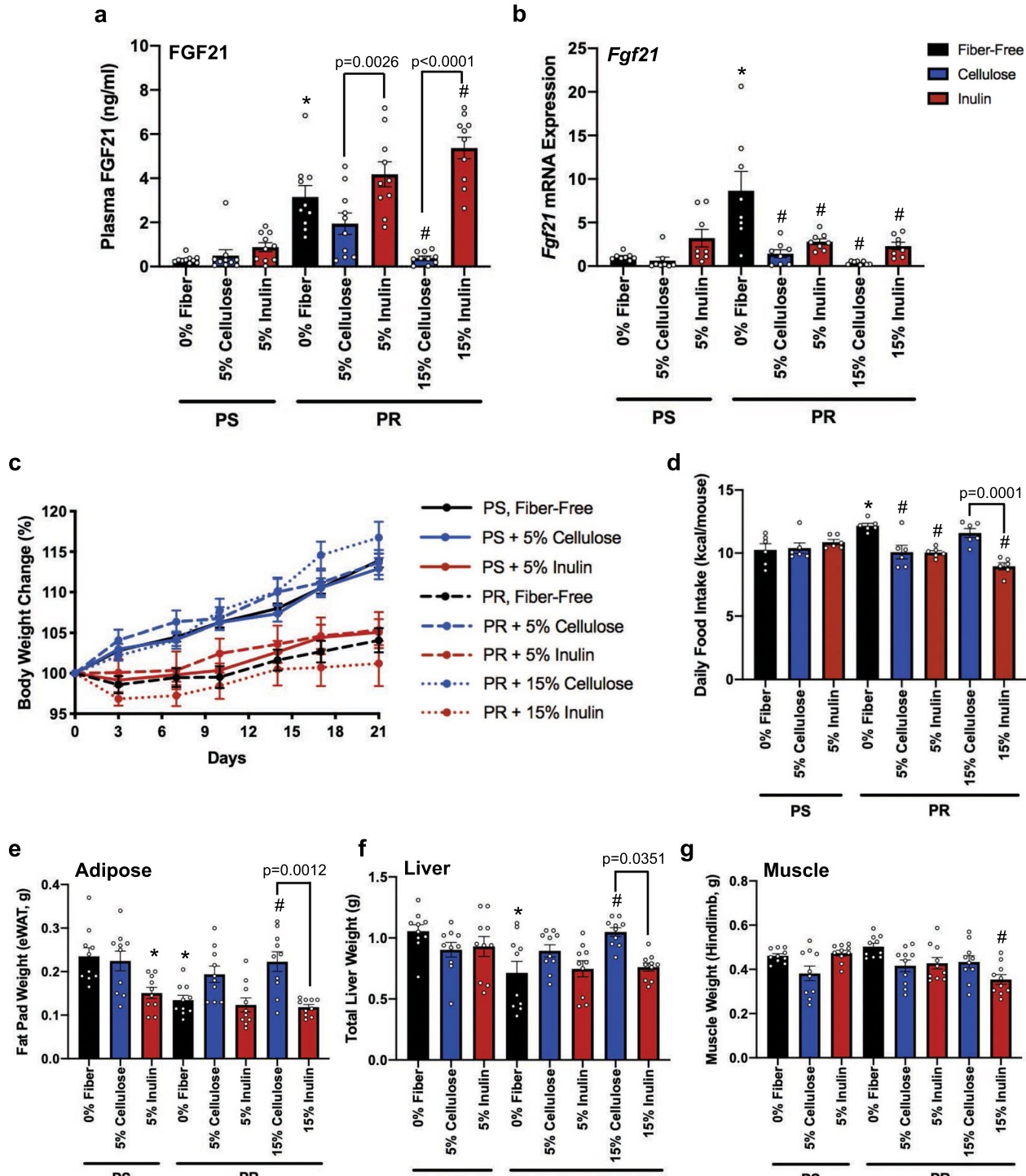

**Fig. 2 Fiber supplementation differentially alters the FGF21 response to a protein-restricted diet.** Mice were fed either protein-sufficient or 10% protein-restricted diets for 21 days. Isocaloric protein-restricted diets were supplemented with 5% or 15% cellulose, or 5% or 15% inulin. **a** Circulating FGF21 protein levels ($n = 10$ biologically independent animals), **b** Hepatic *Fgf21* gene expression measured by qPCR ($n = 8$ biologically independent animals), **c** Body weight change over time ($n = 10$ biologically independent animals), **d** Average kcal consumed per mouse per day. Data points represent per cage average over six-time points. **e** Epididymal fat mass ($n = 10$ biologically independent animals), **f** Liver mass ($n = 10$ biologically independent animals) and, **g** Hindlimb muscle mass ($n = 10$ biologically independent animals). One-way ANOVA was used to determine significance. *$p < 0.05$ = comparison to fiber-free, PS control. #$p < 0.05$ = comparison to fiber-free, PR. All data are represented as mean ± SEM. PS protein sufficient, PR protein restricted.

supplementation strategies that may mitigate potential protein deficiency when simply adding protein back into the diet is not viable. Microbial-accessible fibers such as inulin are typically fermentable in terms of the microbiota's ability to further metabolize it into metabolites for its own, as well as the host's, benefit.

Poorly fermentable fibers such as cellulose, however, typically pass through the GI tract as a bulking agent to promote motility, with little further digestion by the gut microbiota[30]. We hypothesized that of the two tested fibers, inulin would mitigate FGF21 levels because this fiber has greater fermentability by the gut

microbiome than cellulose, and therefore may provide an accessible carbon source for AA or other metabolite generation. However, we observed that when protein was restricted, it was cellulose supplementation that significantly attenuated the FGF21 response whereas inulin did not (Fig. 2a, b). Conversely, when protein was provided in sufficient quantity, fiber supplementation had no impact on FGF21 levels (Fig. 2a, b). We tested whether the cellulose and inulin response on the PR diet was dose-dependent by supplementing both fibers at two different levels, 5 and 15%. The FGF21-mitigating effect of cellulose occurred in a dose-dependent manner while FGF21 remained significantly elevated with inulin supplementation at both levels (Fig. 2a).

The FGF21 status of these mice inversely correlated with their overall growth trajectory during the three-week time course. In the absence of fiber, PR-fed mice exhibited stunted growth in comparison to PS, control mice (Fig. 2c; black lines), despite a significant increase in food consumption (Fig. 2d; black bars). This is consistent with established literature showing that increased FGF21 induces weight loss even when calories are not restricted[16,31,32]. This has been correlated with an increase in energy expenditure[11], however we were unable to measure energy expenditure in our mice. In addition, mice consuming inulin-supplemented PR diets experienced no mitigation of weight loss and were comparable in weight to mice fed a fiber-free, PR diet (Fig. 2c; red lines). Furthermore, the addition of inulin to a PS diet prevented the growth trajectory observed in mice fed a fiber-free, PS diet despite comparable food intake (Fig. 2c, d). Conversely, mice consuming cellulose supplemented PR diets grew comparably to PS diet-fed controls (Fig. 2c; blue lines) and demonstrated similar food intake relative to control (Fig. 2d; blue bars). These disparate growth trajectories in the inulin and cellulose treatment groups were associated with significant differences in both adipose and liver tissue mass (Fig. 2e, f) but not skeletal muscle (Fig. 2g). Specifically, PR resulted in loss of adipose and liver tissue mass, however this was attenuated with cellulose-supplementation. Consistent with the body weight loss in the inulin-supplemented groups, these mice also had diminished fat pad and liver weights. Previous literature has shown that elevated FGF21 tends to result in fat loss[31–35]. Our data are consistent with this, and further demonstrate that inulin supplementation does little to influence this, whereas cellulose supplementation, in the context of PR, attenuates the FGF21 stress response and maintains growth.

**Fiber-supplemented protein-restricted diets differentially alter FGF21-related metabolic pathways**. The differential effect of fiber supplementation on the FGF21 metabolic response to PR diets provides insights into the impact that dietary interventions may play in dampening or sustaining the appropriate adaptive metabolic processes to nutritional stress potentially via the gut microbiota. Before exploring potential gut microbiota roles, we sought to determine if dietary fiber type and amount, on the background of a PR diet, alters associated metabolic pathways related to FGF21 activation. Under nutritional stress, increased hepatic *Fgf21* expression and circulating FGF21 is driven by activation of eIF2α through phosphorylation by GCN2 (Fig. 3a). Our results show that the phosphorylation status of the upstream signal to *Fgf21*, eIF2α, was robustly elevated in the fiber-free PR group in comparison to the fiber-free PS, control group (Fig. 3b), which coincides with activation of this classical pathway. Regardless of fiber type, the addition of either cellulose or inulin to the PR diet suppressed activation of eIF2α (Fig. 3b). However, the degree of eIF2α attenuation was significantly greater in the cellulose group (Fig. 3b), mirroring the trend observed in circulating FGF21 protein levels (Fig. 2a).

In addition to FGF21, we sought to determine if other adaptive processes, such as endogenous AA synthesis, were altered in a compensatory fashion relative to restricted dietary protein intake. We measured hepatic AA biosynthetic genes, asparagine synthetase (*Asns*), and 3-phosphoglycerate dehydrogenase (*3pgd*), which are involved in the production of non-essential AAs asparagine and serine, respectively. As expected, both genes were significantly upregulated in PR mice compared to control, PS mice in the absence of fiber (Fig. 3c, d). The addition of either 5% or 15% cellulose to the PR diet attenuated the expression of both AA biosynthetic genes and were comparable to the fiber-free, PS control (Fig. 3c, d). A similar, but less pronounced response was observed for *3pgd* gene expression with supplementation of either 5% or 15% inulin (Fig. 3d). However, inulin did not attenuate *Asns* as its expression level remained elevated and comparable to the fiber-free, PR diet (Fig. 3c). Furthermore, the addition of inulin to the PS, control diet resulted in a significant increase in *Asns* gene expression (Fig. 3c), which parallels the elevated phosphorylation of eIF2α in this group (Fig. 3b). These data further demonstrate that cellulose-, but not inulin-supplemented PR diets, support normal growth patterns by dampening the nutrient stress pathway.

**Fiber-supplemented protein-restricted diets promote differential shifts in gut microbiota composition**. To understand how fiber supplementation in the context of PR may be changing the gut microbiota to influence FGF21 status, we performed 16S rRNA gene sequencing on stool samples from mice fed each of the aforementioned diets. Principal coordinate analysis revealed a wide spread in β-diversity within both the fiber-free PR and PS groups (Fig. 4a; black circles and triangles), however when 5% cellulose was added to both groups they clustered more closely together (light blue circles and triangles) suggesting that the addition of 5% cellulose can potentially 'normalize' the differences caused by altered protein levels. In contrast. the addition of 5% inulin to the PS and PR groups (light red circles and triangles) maintained clustering by protein status suggesting inulin could do little to mitigate microbiota differences caused by altering protein status. Supplementing PR with a higher level of inulin at 15% did not affect β-diversity as this group clustered with the PR + 5% inulin group (light and dark red circles). However, PR + 15% cellulose clustered distinctly from the PR + 5% cellulose group (light and dark blue circles) and furthest away from the fiber-free, PR group. A similar trend was observed regarding α-diversity whereby fiber-free PS and PR were not different from each other, but the addition of cellulose to both PS and PR, and at any level resulted in increased α-diversity and significantly more so than with inulin supplementation (Fig. 4b).

Differences in bacterial diversity among the various treatment groups were largely explained by whether cellulose or inulin was present in the diet and particularly in the protein-restricted groups. This suggests that on a background of PR, cellulose may actually have the ability to shift the gut microbiota composition despite its consideration as a generally non-fermentable fiber. To understand this better, the microbiota composition was analyzed to determine whether entire classes of bacteria or certain members were responsive to cellulose supplementation. At the phylum level, the primary distinction between PS and PR was largely interindividual differences (Fig. 4c; top panel), which is reflected in the PCoA (Fig. 4a). However, the addition of either fiber type to the PS or PR diet resulted in an overall average increase in Verrucomicrobia (Fig. 4c; top panel), and this is attributed to the genus *Akkermansia* (Fig. 4c; bottom panel). When protein levels are sufficient, inulin appears to promote *Akkermansia* expansion greater than cellulose, however, when

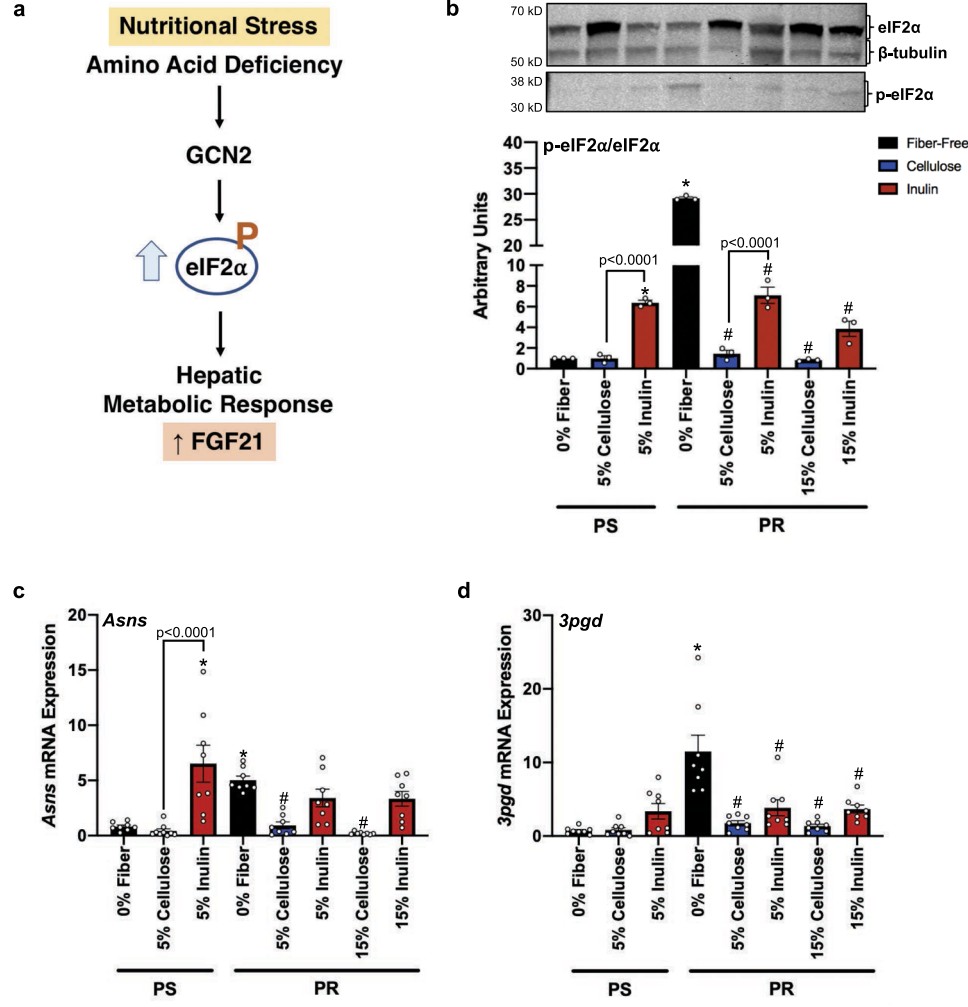

**Fig. 3 FGF21-associated response pathways to protein-restriction are attenuated by fiber supplementation. a** Metabolic stress response pathway to protein deficiency (illustration generated using BioRender). **b** Western blot of hepatic eIF2α phosphorylation ($n = 3$ independent experiments) and, **c-d** Hepatic *Asns* and *3pgd* gene expression measure by qPCR ($n = 8$ biologically independent animals). One-way ANOVA was used to determine significance. *$p < 0.05$ = comparison to fiber-free, PS control. #$p < 0.05$ = comparison to fiber-free, PR. All data are represented as mean ± SEM. PS protein sufficient, PR protein restricted.

protein is restricted, it is cellulose that promotes its expansion. Furthermore, this suggests that fiber effects on the gut microbiota are likely dependent on the context of the background diet. A similar observation was seen with the *Bacteroides* genus whereby its relative abundance increased in both the PS and PR groups when either fiber type was added compared to fiber-free controls (Fig. 4c; bottom panel).

DESeq2 analysis was used to identify significant, differentially abundant genera between treatment groups. Comparison of the PR vs. PS group in the absence of fiber was performed to compare protein status alone. This analysis identified an enrichment in *Oscillospira, Lactobacillus, Ruminococcus, Bacteroides*, and *Dehalobacterium* in the PR group relative to the PS control group (Fig. 4d). Next, we investigated how fiber-mediated alterations in gut microbiota composition might explain why cellulose, and not inulin, attenuates the FGF21 response. Because the PR + 15% cellulose (PR + 15C) group clustered separately from all other fiber-supplemented PR diets (Fig. 4a) and most effectively abrogated the FGF21 response (Fig. 2a), the fiber-free, PR vs. PR + 15C groups were compared. Sequence variants belonging to the genera *Parabacteroides, Akkermansia, Turicibacter, Streptococcus, Adlercreutzia, Lactococcus*, and *Oscillopira* were differentially enriched in PR + 15C relative to the fiber-free, PR group alone (Fig. 4e).

**FGF21 response to protein-restriction is attenuated in the absence of gut microbiota**. These analyses demonstrate that dietary protein status can alter the gut microbiota composition, and can also determine FGF21 levels. Therefore, we asked whether the microbiome is actually required for mediating the FGF21 response in the presence of PR. Utilizing GF mice, we tested the effect of PR in the absence of gut microbiota. These mice were fed either a PS diet or a fiber-free, PR diet similar to the previous diets but further protein restricted (8% by weight) in order to clearly demonstrate whether PR signals host-intrinsic pathways or requires a microbiome mediator. Results from this experiment demonstrate that both circulating FGF21 and liver *Fgf21* gene expression were blunted in GF mice despite PR and were comparable to PS controls (Fig. 5a, b). Endogenous AA biosynthetic (*Asns* and *3pgd* gene expression) responses were also blunted in the absence of gut microbiota (Fig. 5c, d). Furthermore, the weight loss observed in conventional mice fed the PR diet was not seen in GF mice but rather grew within the same trajectory to PS controls (Fig. 5e), likely due to the loss of FGF21 signaling, and was not due to increased food consumption (Fig. 5f). This is similar to the responses reported in FGF21 knock out (KO) mice whereby PR-induced effects on feeding behavior, energy expenditure, body weight gain, metabolic gene expression are

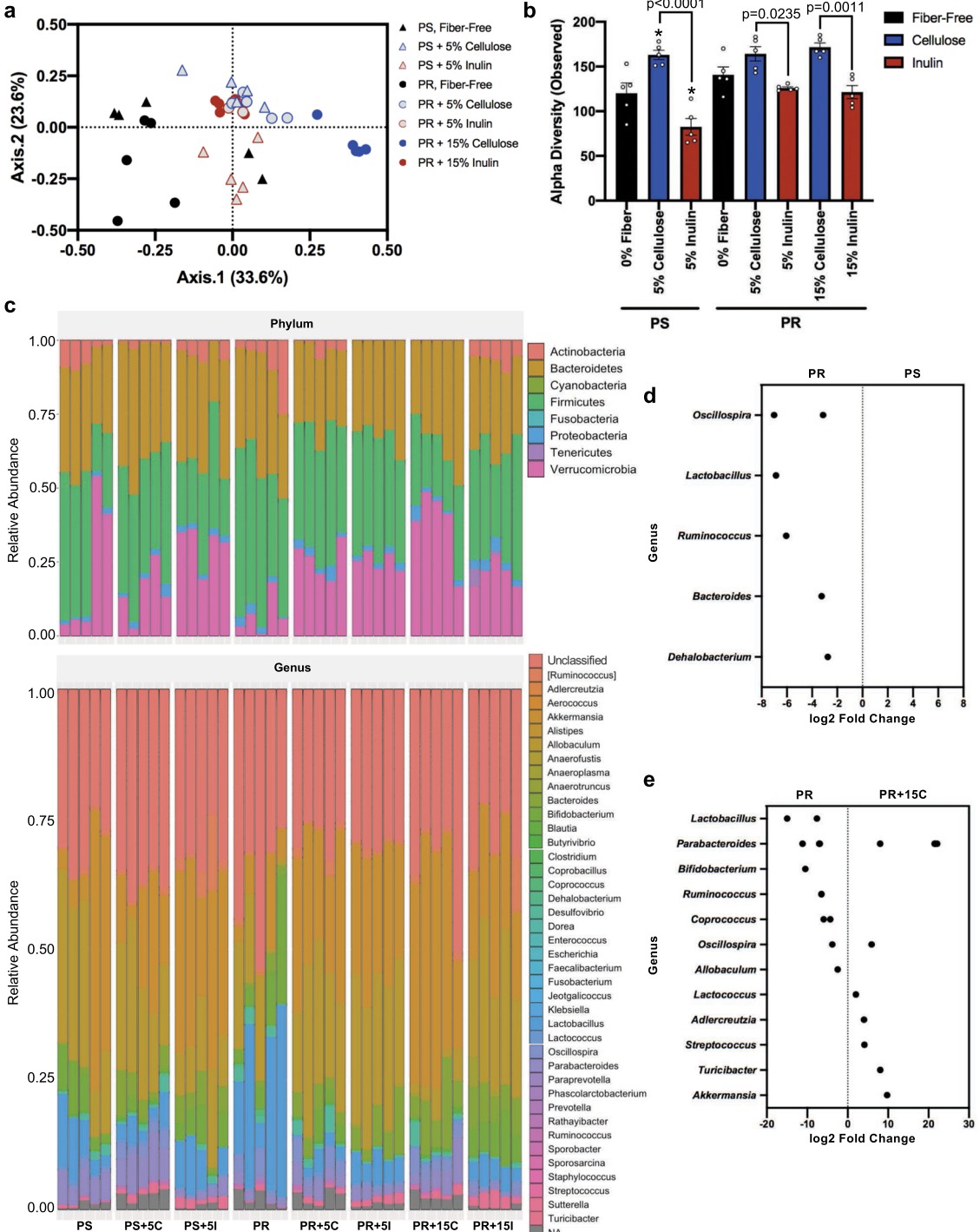

**Fig. 4 Gut microbiota are differentially altered by fiber supplementation.** Amplicon sequence variants were determined from 16S rRNA sequencing of stool experimental endpoint (Week 3). **a** PCoA based on Bray-Curtis distance, **b** Number of unique taxa across diets, **c** Phylum level [top panel] and genus level [bottom panel] relative abundance, **d** Comparison of significant differentially abundant genera using DESeq2 at week 3 in fecal samples from fiber-free, protein-restricted vs. protein-sufficient groups, and **e** Protein restricted vs. protein-restricted + 15% cellulose (PR + 15C). One-way ANOVA was used to determine significance. *p < 0.05 = comparison to fiber-free, PS control. All data are represented from n = 5 biologically independent animals as mean ± SEM. PS protein sufficient, PR protein restricted, C cellulose, I inulin.

completely absent. These FGF21 KO mice grow normally in the face of PR, and more so than their wild-type counterparts on PR[36]. If the absence of gut microbiota can create the same degree of FGF21 desensitization to PR as FGF21 KO mice, this suggests that gut microbiota signaling is an integral part of the host nutrient stress sensing pathway.

**Dietary fibers promote compositional shifts in gut microbiota that selectively influence the FGF21 stress response.** Given that the gut microbiome appears to be a necessary component of PR-induced FGF21 signaling, we aimed to determine which members of the microbiota that were most altered when fiber was added to the PR diet were also correlated with attenuating FGF21 levels.

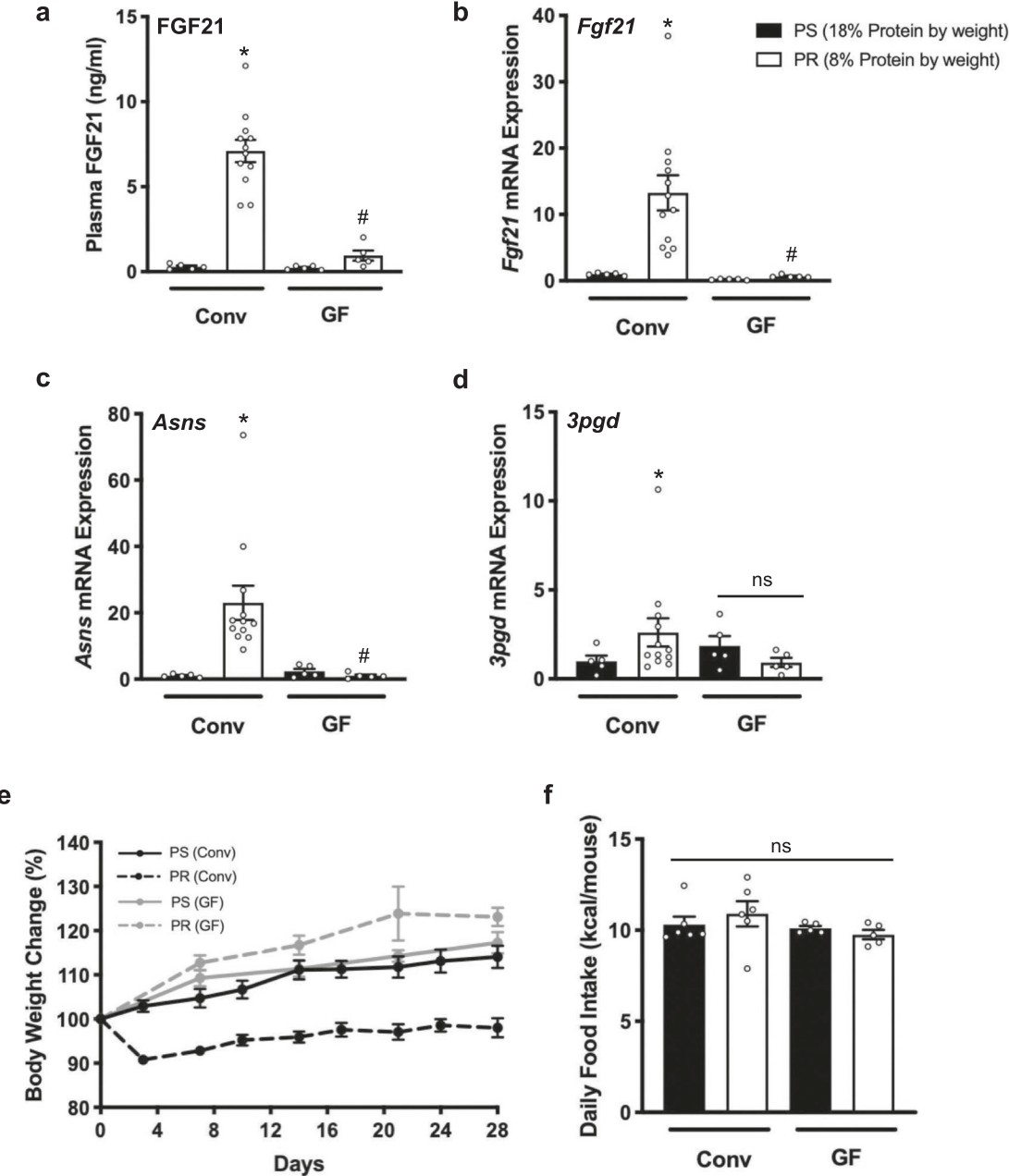

**Fig. 5 Gut microbiota influence the FGF21 response to PR.** Conventional (Conv) and germ-free (GF) mice were fed either a protein-sufficient or a protein-restricted diet (8% protein by weight) for 28 days. **a** Circulating FGF21 protein levels, **b**–**d** Hepatic *Fgf21, Asns*, and *3pgd* gene expression measured by qPCR, respectively, **e** Body weight change over time and, **f** Average kcal consumed per mouse per day. Data points represent per cage average over six-time points. One-way ANOVA was used to determine significance. *$p < 0.05 =$ comparison to Conv mice fed a PS diet. #$p < 0.05 =$ comparison to Conv mice fed a fiber-free, PR. All data are represented as mean ± SEM. Conv-PS, GF-PS, and GF-PR represents $n = 5$ biologically independent animals; Conv-PR represents $n = 12$ biologically independent animals. PS protein sufficient, PR protein restricted.

We found that *Sutterella* was the only genus whose relative abundance positively correlated with plasma FGF21 concentrations while *Parabacteroides, Lactococcus, Blautia, Streptococcus*, and *Anaerofustis* were negatively correlated with FGF21 (Fig. 6a). *Parabacteroides, Lactococcus*, and *Blautia* were also negatively correlated with both *Asns* and *3pgd* gene expression while *Streptococcus* was only negatively correlated with *Asns* gene expression (Fig. 6a). Of these genera, *Parabacteroides, Lactococcus*, and *Streptococcus* were differentially and significantly enriched in the PR + 15C group relative to the fiber-free, PR group as demonstrated in the DESeq2 analysis (Fig. 4e). Furthermore, a comparison of differentially abundant taxa between a PR diet supplemented with 15% inulin (PR + 15I) and PR + 15C revealed that FGF21 negatively correlated with genera, *Parabacteroides, Lactococcus, Blautia*, and *Streptococcus*. These were all enriched in the PR + 15C group while *Sutterella*, which was positively correlated with FGF21, was enriched in the PR + 15I group (Supplementary Fig. 1a). A similar trend was paralleled at 5% supplementation when comparing PR + 5% cellulose (PR + 5C) and PR + 5% inulin (PR + 5I) whereby *Parabacteroides* and *Lactococcus* were enriched in PR + 5C while *Sutterella* was enriched in PR + 5I. (Supplementary Fig. 1b). Therefore, these three genera (*Parabacteroides, Lactococcus*, and *Sutterella*) emerge as candidate

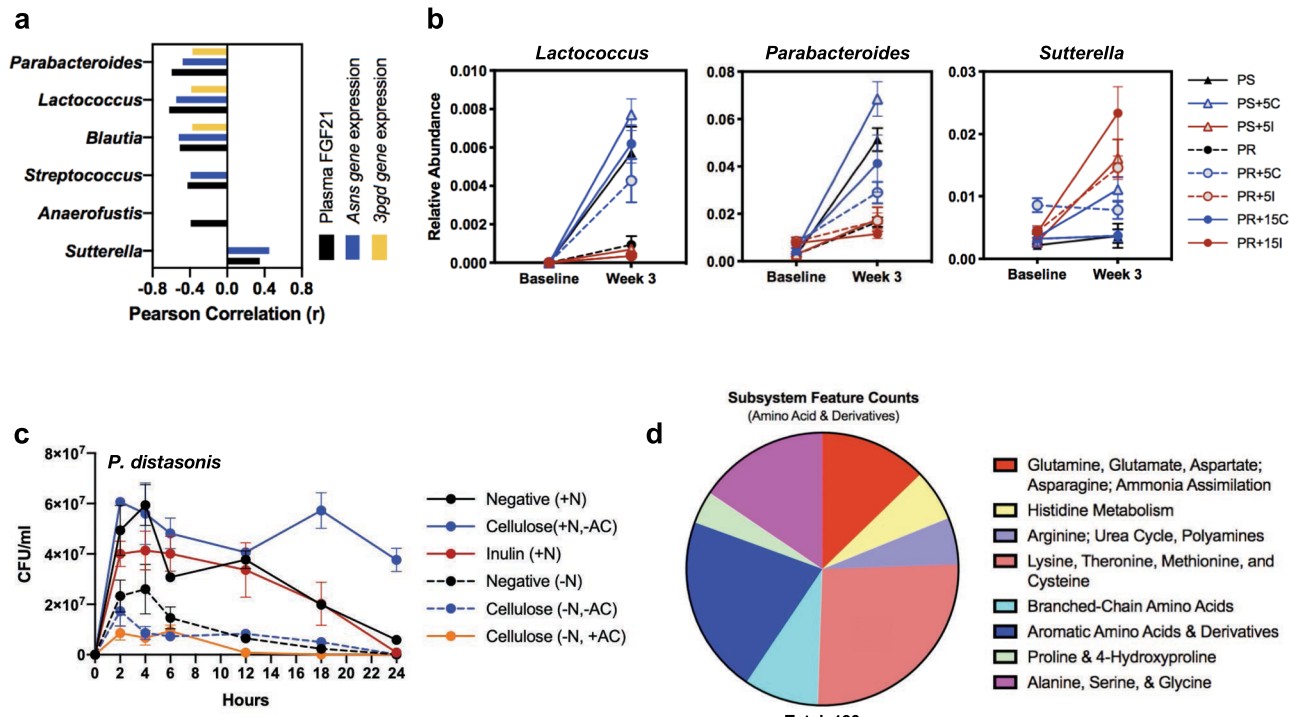

**Fig. 6 Cellulose responsive gut microbiota negatively correlate with FGF21 status. a** Significant Pearson correlations between FGF21 plasma concentration, *Asns*, or *3pgd* gene expression and relative abundance of bacterial genera, **b** In vivo change in bacterial relative abundance in mice at baseline versus week 3. Data are represented by *n* = 5 biologically independent mice as mean ± SEM, **c** In vitro cultivation of *P. distasonis* in minimal media supplemented with ±NH₄Cl (N), cellulose, or inulin. Data are represented by *n* = 3 independent experiments as mean ± SEM, **d** AA subsystem feature counts from whole-genome sequencing of *P. distasonis*. PS protein sufficient, PR protein restricted, AC autoclave, C cellulose, I inulin.

groups that might be influencing the FGF21 response to PR when supported by fiber supplementation.

The relative abundance of these three genera was compared at baseline to experimental endpoint (Week 3) across diet treatments to determine if there was a clear pattern in fiber preference between these bacteria. This analysis revealed that cellulose supplementation, regardless of protein status, promoted the growth of both *Parabacteroides* and *Lactococcus* more so than inulin (Fig. 6b). In the absence of fiber, *Parabacteroides* and *Lactococcus* grew more robustly in mice fed PS rather than PR diets, but was further enhanced by the addition of cellulose (Fig. 6b). These data coincide with our finding that *Parabacteroides* and *Lactococcus* negatively correlated with FGF21, as FGF21 levels were reduced in the fiber-free, PS group as well as any group supplemented with cellulose (Figs. 2a and 6a, b). Conversely, *Sutterella* was enriched at experimental endpoint when mice were fed diets containing inulin regardless of protein amount (Fig. 6b). This trend also parallels our data indicating that *Sutterella* positively correlates with FGF21 status which was elevated in inulin-supplemented PR diets (Fig. 2a).

The above findings showing a robust effect of cellulose in modulating the gut microbiota during PR is intriguing given that most cellulose is considered non-fermentable by human and murine enteric bacteria. However, at the species-level, our differential abundance analysis revealed that sequence variants belonging to the *Parabacteroides* genus that were significantly enriched in both PR + 15C and PR + 5C mice in comparison to PR + 15I and PR + 5I, respectively, were classified as *Parabacteroides distasonis* (Supplementary Fig. 1a, b). We isolated *P. distasonis* from the stool of the PR + 15C group, confirmed by full-length 16S rRNA gene sequencing, and performed an in vitro growth curve assay to determine if *P. distasonis* could preferentially grow in cellulose enriched minimal media as

opposed to inulin. In addition, we tested whether the presence or absence of nitrogen in the media would affect the growth of this isolate to determine the hierarchy of substrate requirements for *P. distasonis*. For the first twelve hours of the time course, *P. distasonis* grew comparably in minimal media alone or with cellulose or inulin supplement (Fig. 6c; solid lines). However, viable *P. distasonis* counts decreased in both minimal media alone and with inulin supplement while cellulose supplementation sustained survival of the bacterium through the end of the time course. We also tested the cellulose-supplemented media with and without autoclaving to determine whether heat was inducing any inadvertent liberation of a carbohydrate nutrient source from the media. We found no difference in growth with and without autoclaving (Fig. 6c, solid orange line and dotted blue line). Furthermore, we determined that the sustained growth of *P. distasonis* on cellulose supplemented media was completely inhibited when media did not contain a nitrogen source (Fig. 6c). Therefore, a nitrogen source is required for *P. distasonis* growth, and when that requirement is met, growth capacity and fitness of *P. distasonis* is further enhanced in the presence of cellulose over inulin.

Because *P. distasonis* demonstrates an ability to grow robustly in environments containing cellulose, we explored how *P. distasonis* might contribute to the mitigation of the FGF21 stress response. We performed whole-genome sequencing on our murine-derived *P. distasonis* isolate and found that the top three functional genomic categories were associated with (1) carbohydrates, (2) AAs and their derivatives, and (3) protein metabolism, accounting for 17.0, 14.4, and 13.9%, respectively, of the total feature counts (Supplementary Fig. 2). Within the carbohydrate category, there were several features that mapped to hydrolytic enzymes (i.e., glucosidases and xylanase) responsible for breaking down complex oligosaccharides such as hemi-cellulose.

Furthermore, within the AAs and derivatives category, we discovered that *P. distasonis* has the genetic capacity to de novo synthesize essential amino acids (EAAs) including histidine, lysine, methionine, and threonine (Fig. 6d, Supplementary Table 1). Taken together, these data demonstrate that cellulose-based fiber supplementation may mitigate protein deficiency through a mechanism that involves the gut microbiota.

## Discussion

The survival and fitness of symbiotic gut microbiota and the mammalian host rely on a mutualistic paradigm of "commensal commerce" defined by the bidirectional exchange of resources to promote metabolic homeostasis. Therefore, alterations in host-microbe interactions can occur as a result of prolonged intake of nutritionally imbalanced diets. Our study suggests that manipulations of the gut microbiota by fibers are more nuanced than previously thought and largely dependent on the background diet- in this case PR. For instance, Zou et al.[37] demonstrate that inulin and not cellulose improves metabolic readouts related to insulin signaling and hepatic steatosis, via changes in the gut microbiome, but these studies have been on backgrounds of high-fat diets.

On a background of dietary PR, we confirm previous reports that PR induces a highly reproducible hepatic FGF21 response that is mirrored in elevated plasma FGF21 levels. However, we additionally demonstrate that a protein-restricted diet supplemented with cellulose resulted in attenuation of the FGF21-mediated host nutritional stress pathway to the point of appearing no different from protein-sufficient controls. Gut microbiota compositional analyses revealed that cellulose supplementation skews the gut microbiota toward a composition that particularly favors two species in particular whose increases in relative abundance are negatively correlated to FGF21 levels in the host. In other words, when the abundance of these organisms are increased in the presence of cellulose, the host does not appear to sense the nutrient stress of PR. We also discovered that FGF21 is non-responsive in protein-restricted GF mice, demonstrating a requirement for the gut microbiome in mediating this nutritional stress response. When the gut microbiota are present, FGF21 signals appropriately in response to PR, as demonstrated in conventionally raised mice. So in the absence of a gut microbiota, the FGF21 response to a PR diet is lost and animals grow normally despite decreased protein. However, we find this can also be achieved with an intact microbiome if cellulose is supplemented. This suggests there is a microbially produced signaling molecule, whether it is free amino acid production or other microbial metabolite, that is mitigating FGF21. FGF21 is the primary endocrine molecule in the host that senses these dietary AA imbalances, and this study demonstrates that FGF21 may be influenced by the activities of the gut microbiome. The human and murine genomes do not possess genes capable of cellulose degradation, therefore the observed effect of cellulose supplementation altering host FGF21 levels indicated the most plausible mediator is the gut microbiota. Indeed, we found that several members of the gut microbiota increased in relative abundance with cellulose supplementation and specifically, *P. distasonis* was capable of utilizing cellulose when tested in vitro. Whole-genome sequencing also revealed that this bacterium also has the ability to generate EAAs presuming sufficient carbon and nitrogen substrate is available.

A working hypothesis may be that the cellulose-specific enrichment of bacterial species (such as *P. distasonis*) capable of de novo synthesizing EAAs during states of prolonged PR produces sufficient amounts of EAAs for host utilization to compensate for their limitation in the diet. This, in turn, would

blunt the hepatic FGF21 stress response to PR. Indeed, recently, Quinn et al. discovered novel gut microbiota synthesized AAs conjugated to host-derived bile acids including two EAA conjugates, phenylalanocholic acid and leucocholic acid, which may suggest a possible mode of delivery for microbially-generated AAs in the gut for host uptake[38]. This line of investigation could improve dietary recommendations for those consuming low-protein diets, and furthermore provide insight into delaying sarcopenia and treating kwashiorkor.

## Methods

**Animals and diets**. C57BL/6 conventionally raised mice were housed at Cedars-Sinai Medical Center, Los Angeles, CA, USA and bred for seven generations prior to use. The mice were housed in a 12-h light: dark cycle at 22 °C–25 °C temperature with relative humidity of 50–70%. All experiments herein were approved and comply to ethical guidelines for animal testing and research. The animal protocol used in this study was approved by the Cedars-Sinai Institutional Animal Care and Use Committee. Male mice at 6–8 weeks of age were fed one of eight diets (Envigo): either a fiber-free protein-sufficient control diet (18% by weight) diet based on AIN-93G diet, or a protein-sufficient diet supplemented with either 5% cellulose or 5% inulin, or a fiber-free protein-restricted diet (10% by weight), or a protein-restricted diet supplemented with either cellulose or inulin with the following doses: 5% or 15% for each fiber type (Envigo, Madison, Wisconsin) for 3 weeks ($n = 10$/group). Diet formulations are included in Supplementary Table 1. Food intake and body weight were measured every 3 days. Stool, blood, liver, muscle, and fat were collected for downstream analyses. For GF mouse experiments, eight-week old male GF ($n = 5$) C57Bl6/J mice were obtained from Jackson laboratory. The mice were housed in the GF facility in a 12-h light: dark cycle at 20 °C–22 °C temperature with relative humidity of 30–70% at the University of California, San Diego, CA, USA and allowed to acclimate for 14 days with ad libitum access to an irradiated protein-sufficient control (18% by weight, Envigo). Mice were then switched and fed ad libitum on an irradiated protein-restricted diet (8% by weight; Envigo). Mice were individually housed, body weight, and food intake were measured weekly throughout the 28-day experimental period ($n = 5$/group). At sacrifice, blood and liver samples were harvested for downstream analyses. These experiments were approved and comply to ethical guidelines for animal testing and research. The animal protocol used in this study was approved by University of California, San Diego Institutional Animal Care and Use Committee. Diet formulations are included in Supplementary Table 2.

**RNA isolation and quantitative RT-PCR**. Total RNA was extracted from the liver using TRIzol (Invitrogen, Carlsbad, California) according to the manufacturer's protocol, and reverse transcribed using the iScript One-Step RT-PCR Kit with SYBR Green (Bio-Rad, Hercules, California). Primers are listed in Supplementary Table 3. Differences in transcript levels were quantified by normalization of each amplicon to housekeeping gene *Rps17*.

**DNA extraction and 16S rRNA sequence analysis**. DNA was extracted from baseline and week three stool samples using the DNeasy PowerSoil Kit (Qiagen, Germantown, MD) with additional steps to maximize cell lysis. Stool was added to lysis tubes with 400 ug proteinase K (Invitrogen) and homogenized at 5 m/s for 2 min. This was followed by heat treatment at 95 °C for 15 min and centrifugation at $16,000 \times g$ for 5 min at 4 °C. The supernatant was transferred to a new tube and reserved for later use. Three hundred microliters of fresh lysis buffer was added back to the lysis tube for the second round of homogenization and heating. Supernatant from both rounds of cell lysis were pooled for DNA isolation as per the manufacturer's protocol. The V4 region of the 16S rRNA gene were sequenced on the MiSeq Platform (Illumina, San Diego, CA). Reads were quality filtered, trimmed, merged, denoised, chimera filtered, and binned into sequence variants using DADA2 v1.14.1. Sequence variants were aligned to the GreenGenes reference database v13.8.

**FGF21 ELISA**. Plasma was isolated from whole blood centrifuged at $3000 \times g$ for 15 min at 4 °C. FGF21 concentration was measured using the Mouse and Rat FGF-21 ELISA Kit (Biovendor, Brno, Czech Republic) following the manufacturer's protocol.

**Western blot analysis**. Liver tissues were hand homogenized by mortar and pestle in 5 mL of protein lysis buffer: T-PER™ and 100X Halt Protease Inhibitor and Phosphatase Inhibitor (Thermo Fisher, Asheville, NC). Protein concentration was determined using the Pierce™ Bradford Assay (Thermo Fisher Scientific, Asheville, NC). Gel electrophoresis was performed on a 10% polyacrylamide gel. Proteins were transferred onto PVDF membranes and blocked using TBS Odyssey Blocking Buffer (Li-Cor, Lincoln, NE). Membranes were incubated overnight at 4 °C in Odyssey Blocking Buffer with primary antibodies. Membranes were washed using 1X TBS (Li-Cor, Lincoln, NE) containing 10% Tween 20 (Sigma-Aldrich, St. Louis,

MO) followed by incubation with the secondary antibodies. Blots were imaged on the Odyssey Classic (Li-Cor, Lincoln, NE), and bands were scanned and quantified using Image Studio Lite (Li-Cor, Lincoln, NE).

The following primary antibodies were used: Anti-eIF2α (Abcam, Cambridge, MA; Cat # ab169528, Lot # GR321237-4; 1:1000 dilution), Anti-phospho-eIF2α (Cell Signaling, Danvers, MA; Cat # 3398 S, Lot # 6, 1:500 dilution), and Anti-β-Tubulin (Sigma Aldrich, St. Louis, MO, Cat # T4026, Lot # 125M4884V; 1:200 dilution).

**Bacterial cultivation**. Freshly collected stool pellets from mice fed protein-restricted diet + 15% cellulose were immediately brought into an anaerobic chamber and homogenized in pre-reduced PBS to isolate bacteria with potential cellulose-degrading capacity. Homogenate was serially diluted and spread plated on chocolate blood agar (Becton Dickson, Franklin Lakes, NJ) and incubated at 37 °C. Distinct colonies were re-streaked on day 4 and 7 post-incubation. Colony PCR was used to identify the isolated organisms. Briefly, cells were scraped from culture media and then lysed in sterile water at 97 °C for 20 min. Lysates were pelleted in mini-centrifuge, and the supernatant served as template in PCR. Primers 27F and 1492R (Supplementary Table 3) and the iTaq DNA polymerase kit (Bio-Rad, Hercules, CA) were used to amplify the 16S rRNA gene. Amplicons were submitted to Laragen for Sanger sequencing, and the resultant sequence traces were visualized in FinchTV v1.4. The trimmed reads were identified by Microbial BLAST.

**Bacterial growth curve assay**. *P. distasonis* was resuspended in pre-reduced PBS and normalized to an $OD_{600}$ reading of 0.2. In a 96-well format, 175 ul of each media was added to the plate and inoculated with 25 ul of normalized *P. distasonis*. During the time course, *P. distasonis* was incubated under anaerobic conditions. Minimal media contained 5 g/L $KH_2PO_4$, 12.8 g/L $Na_2HPO_4 \cdot 7H_2O$, 0.5 g/L NaCl and was supplemented with or without 1.0 g/L $NH_4Cl$. In addition, cellulose or inulin (Sigma-Aldrich, St. Louis, MO) were added at a final concentration of 0.4%. At the respective time points, 50 ul of the inoculated media was plated onto chocolate blood agar plates and incubated in an anaerobic chamber for 48 h prior to quantification of CFUs.

**Whole-genome sequencing and analysis**. Genomic content from *P. distasonis* isolate was extracted from scraped colonies using the DNeasy PowerSoil Kit (Qiagen, Germantown, MD) as per manufacturer's protocol. Purified DNA was sent to the Microbial Genome Sequencing Center at the University of Pittsburgh for library preparation, followed by whole-genome sequencing on the Illumina NextSeq 550 flow cell. The raw paired-end reads were quality trimmed using the Nesoni pipeline. The quality-filtered reads were assembled into a 5.53 Mb genome with 53 contigs and 45.2% G + C content using *SPAdes assembler*[39] at a k-mer length set between 41 and 61. The contigs were then annotated into broad pathway level (i.e., subsystems) and enzyme level features using RAST annotation server[40] for functional roles. For the purpose of the study, functional annotations were further parsed for the amino acid metabolism associated enzymes which were then mapped to corresponding pathways using EC numbers.

**Quantification and statistical analysis**. Statistical significance was determined by one-way ANOVA with Tukey or Dunnett post hoc test for multiple comparisons or by student's *t*-test for comparisons between two groups. Microbiome sequences were subsampled to 20,000 reads per sample prior to diversity assessments. α-diversity (i.e., bacterial variety and complexity within a sample) and β-diversity (differences in composition across diets) were calculated using the phyloseq package v1.30 in R version 3.6.2. Community difference was assessed using Bray-Curtis distance and visualized by principal coordinate analysis. DESeq2 v1.26 was used to test for significant differentially abundant sequence variants between diets. The relationship between plasma FGF21 and change in community composition was assessed with Pearson's correlation test in GraphPad Prism v.8.2.1. A threshold of $p < 0.05$ was considered statistically significant.

**Reporting summary**. Further information on research design is available in the Nature Research Reporting Summary linked to this article.

## Data availability

All data generated or analyzed during this study are included in this published article and its supplementary tables. In addition, source data are provided with this paper. The 16S rRNA gene and bacterial whole-genome sequencing data have been made publicly available through the NCBI SRA database with accession numbers PRJNA726200 and PRJNA726208 respectively. Databases used in this study are as follows: Microbial BLAST (https://blast.ncbi.nlm.nih.gov/Blast.cgi?PAGE_TYPE=BlastSearch&BLAST_SPEC=MicrobialGenomes); RAST Server-PATRIC (https://rast.nmpdr.org); Greengenes v13.8 (https://greengenes.secondgenome.com/). Source data are provided with this paper.

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

## Acknowledgements

We would like to thank Dr. Katie Markan, and Dr. Janelle Ayers for their guidance and suggestions.

## Author contributions

A.M., S.D., G.E.M., and J.G., designed experiments, A.M., S.D., and D.K.L designed diet formulations. A.M., G.E.M., C.W.Y.H., and G.H. performed experiments. All authors were involved in analysis of data. A.M., D.K.L., J.G., and S.D. prepared and wrote the manuscript with input from all other co-authors.

## Competing interests

The authors declare no competing interests.
