## [Peer Review File · Nature Communications]

Reviewers' Comments:

Reviewer #1:

Remarks to the Author:

I think the basic observations here are interesting but don't think it is quite ready for publication. Several aspects of the experimental design are not very clear and it is hard to see how the various factors known to influence FGF21 signaling were experimentally accounted for. As a consequence, the support for the conclusions at present is not sufficiently robust. I don't think there is much to support the claim in the last line of the abstract and the sentence on L38 to L41 is unsupported in two ways – there is no objective way to categorize the microbiota as 'favorable' and there is no meaningful characterization of how it has changed provided.

The experimental diets need to be more clearly described to understand the rationale and experimental tests. The two basic diets are <10% (specific value should be given) and 18% protein. These are stated to be isocaloric, which presumably means that protein has been replaced with digestible carbohydrates? The experimental test is to examine the impact of fiber supplementation on the outcomes of protein restriction. To this end the two base diets were supplemented with different doses of fiber sources - either inulin or cellulose. Were these fiber-supplemented diets considered isocaloric with the non-supplemented diets? Not so in terms of energy bioavailability.

Inulin is readily microbially digestible and so expected to yield ca 2.5 cal/g to the mouse, whereas cellulose is of limited digestibility and typically inferred to be close to 0. If the diets were kept isocaloric as the text implies then other components must also be adjusted - what were these?

On L57 - L63 (Fig 1) the supplementations are described as being at 3.5% and 10.5% (presumably by weight). On L97-98 (Fig 2) the supplementations are at 5% and 15%. Clarification of why the change was made should be given.

It is surprising that the mice did not increase food consumption in response to protein dilution (Fig. 1b), as has been reported many times previously. This needs to be discussed somewhere.

L188 - 196 The change in relative abundance should not be described as a bloom, unless you have evidence that it is due to growth. The growth on M9 with cellulose as sole carbon source is interesting. Was the cellulose added to the medium before autoclaving and if so did you exclude the possibility of growth on sugars released during sterilization? A simple experiment here might be testing the ability of *E. coli* to grow on the same medium.

The diet supplementations do not show a clear effect of fiber supplementation in Fig 3. By far the majority of explained variation is due to the transition from baseline to 3 weeks experimental diet. Within the experimental diets there is limited variation but to see this you really need to show a separate ordination at just week 3 to see what is happening.

Fig 4c shows rather modest evidence for growth. The viable counts for *Lactobacillus* are exceedingly low. A ten-fold change in cfu/ml could easily be growth on trace carbon sources either introduced with the inulin or released during autoclaving. Fig 4e is a little more convincing with a 100-fold change but this is still about 4 orders of magnitude less than what would typically expect for a bacterial growth experiment. Ideally this would be a growth curve, and in any event some details on the time of incubation and initial inoculum are needed to interpret.

Reviewer #2:

Remarks to the Author:

This manuscript by Martin and colleagues describes the interesting observation that the induction of FGF21 in response to dietary protein restriction is attenuated when the PR diet is supplemented with cellulose but not inulin, and that these changes in FGF21 secretion are associated with unique changes in the gut microbiota. In addition, the ability of PR to increase FGF21, and thus cellulose to reduce it, are lost in mice with antibiotic based depletion of the gut microbiome. Collectively the

data lead to the interesting conclusion that the response to a protein restricting diet, at least in terms of FGF21 production, somehow depends on an interaction between protein content, dietary fiber, and gut microbiota.

The strength of the manuscript is clearly the novelty of this interaction between FGF21, fiber and the gut microbiome. While a number of studies have clearly demonstrated that protein restriction increases FGF21, this link to the microbiome appears to be unique.

The primary weakness of the manuscript is that the work describes a phenomenon that is largely unexplained. We have little insight into the mechanism through which cellulose lowers FGF21 but inulin does not, the mechanism through which antibiotic based depletion of the microbiome blocks the effect of PR on FGF21, or the larger implications of this work to metabolic responses to PR or dietary fiber. There are also concerns related to design and interpretation. These issues are discussed in more detail in the specific comments below

Specific Comments

1. The diet compositions are not completely clear. The methods reference Table S1 for dietary composition, but Table S1 is the primer sequences. Thus the manuscript does not actually describe the diets in detail, and this leads to confusion that permeates the remaining comments.

2. It is not clear what the appropriate control is for these studies, and whether some of the experiments (Figs 2 and 3) include an appropriate control. There are several related issues:
a. The studies in Figs 2 or 3 lack a PR alone group, yet in Fig 1 and 4 there is a PR alone group. This inconsistency is awkward and raises the concern that there is a missing control (PR alone) in Figs 2 and 3. A key conclusion is that the addition of cellulose blocks the effect of PR. Yet without a PR alone group there is no way to compare the effect of added cellulose or inulin. For instance, in Fig 2 there is no direct evidence that PR increases eIF2a. Second, the result section says eIF2a phosphorylation was 'suppressed by cellulose', but it is impossible to make that conclusion without a PR alone group because there is nothing for it to be suppressed relative to. One can only say it is similar to control. It is also not clear if inulin is producing unique effects, for instance exerting an independent effect to increase FGF21, or if it is simply not having any effect and thus FGF21 is increased solely due to PR. Finally, it noteworthy that none of the experiments test whether cellulose or inulin exert effects in the absence of PR. I would not expect them to, but the current data never address that possibility or include these group control+cellulose or control+inulin groups. Although unlikely, it seems possible that inulin supplementation to a control diet alone might be sufficient to increase FGF21, and if so this would greatly cloud the interpretation.

b. More generally, it is not clear what the appropriate controls should be. The Envigo AIN-93G diet, which I think but am not certain (see point #1) is the control, contains 5% cellulose. Thus the control and PR alone diet (which is fiber-free) differ not just in protein content, but also in cellulose content. Is it possible that the increase in FGF21 with the PR alone diet (Fig 1) is partly due to the reduction in cellulose? Should the control diet (or a control diet) also have zero cellulose like the PR diet? Conversely, one of the diets is PR plus 5% cellulose. If the control has 5% cellulose, then one could argue that this diet is the true PR alone group and not the fiber-free PR diet. Particularly when one considers that the PR diet and the PR+3.5% cellulose diet in Fig 1 do not produce the same effects. The manuscript at least needs to be clearer about the fiber content in the control (possibly labeling it), and more directly deal with these issues of variance in fiber content and appropriate control. Ideally, if a 5% fiber control is used then the PR alone diet should also be 5% fiber and all comparisons should be added fiber beyond this number. Alternatively, a fiber-free control and fiber-free PR group used, and then cellulose or inulin added to these diets. It is also possible that I am mistaken regarding the actual control since it is not directly described, and if so then many of my subsequent comments might be off base. Regardless, this issue needs to be clarified.

c. There is inconsistency in the dietary fiber content and endpoints measured. Figure 1 compares the effects of a 0%, 3.5% and 10.5% fiber, while figure 2 compares only 5% and 15%. It is not clear that 3.5% and 5% fiber are the same, or that 10.5 and 15% is the same. Figure 1 measures physiological endpoints, while Figure 2 does not. Figure 2 measures liver molecular endpoints, which Fig 1 does not. It is difficult to get a complete picture of what is happening due to the

inconsistencies in diets used. It not clear why all the experiments did not use the same diets so that direct comparison between the various figures could be more easily made.

d. In summary, the point of these comments is that there are multiple moving parts among these diets and inconsistency between experiments in the diets used. The diets need to be designed so that either protein or fiber is manipulated/compared, but not both at the same time. The diets need to appropriately consider and articulate the fiber content of the controls, and appropriate controls included in all studies.

3. Does PR itself alter the microbiota? Without a PR alone group, it is impossible to address this question, and therefore it is unclear if the addition of cellulose or inulin is reversing an effect of PR on the microbiota, or instead doing something unique to the microbiota that then exerts downstream effects on FGF21.

4. In Figure 2 there is some discrepancy between changes in FGF21 and changes in ASNS/3pgd, specifically for the 5% cellulose PR group. These latter genes are markers of the liver response to protein restriction, yet neither go up with either cellulose group. As noted above, there is some concern that the Control group also has 5% cellulose, and therefore one wonders if the 5% cellulose group is the actual PR control. But if that were true there should have been an increase in 5% cellulose group. Thus one wonders if the increase in ASNS/3PDG (which are not measured in Fig 1 but should have been), are being driven by the addition of inulin, and not by the exposure to PR? As I see it, there is no way the figure can argue against this conclusion without a PR alone group.

5. I am not sure I agree with the initial discussion of Figure 3. This discussion focuses primarily on the inulin groups, and indeed the title of this section (line 116) references inulin. However, when I focus on Figure 3A the primary observation is the unique clustering of the 15% cellulose group. This group seems more divergent from control than inulin, or at least similarly divergent (but in a different direction). Considering this it is this 15% cellulose group that exerts the unique effect of blocking PR, why isn't this group the primary emphasis? Perhaps I am missing something.

6. Fig 4: The only in vivo physiological endpoint measured in Figure 4 is FGF21. The manuscript would be much stronger if additional endpoints related to hepatic amino acid sensing (eIF2a, ASNS, etc) were measured, as well as whole animal measures of the larger metabolic response to PR. Currently there is are several important unanswered questions: 1) Is the effect of microbiome depletion specific to FGF21, or a more general effect on the liver sensing of PR? 2) Does the depletion of the microbiome also block changes in body weight, etc that occur in response to PR, and are these related to changes in FGF21? It seems very insufficient to just measure FGF21. More generally, it would be nice to simply see the metabolic/bodyweight state of the depleted mice, to get some insight into their health.

7. A major concern of the manuscript is that the effects on FGF21 are largely phenomenological. We have very little insight into what the cellulose is specifically doing that would rescue liver FGF21 production. We also have very little insight into the mechanism through which the depletion of the microbiome blocks the increase in FGF21 during PR. Are these even the same mechanism? Some of these questions might be answered by more thoroughly testing and phenotyping the studies already conducted as noted above. There are several obvious possibilities, but one very important one which I think should be addressed in this manuscript is the possibility that the microbiome is synthesizing amino acids, which are absorbed and detected by the liver, thus preventing the liver response to protein restriction (increased FGF21, ASNS, 3PDG). This question would be partially tested by measuring ASNS/3PDG in the mice in Figure 1 and Figure 4, and as noted above by including appropriate controls in Figure 2. In addition, it would seem possible, but more difficult, to actually measure microbial amino acid production and/or absorption. In other words, is the addition of cellulose preventing the liver from sensing protein restriction (ASNS not increasing, for instance). Or does the liver sense the PR normally, but there is something unique occurring to FGF21?

8. There is a disconnect of sorts between the data in Figure 1-3 and that of Figure 4. The first several experiments suggest that the addition of cellulose blunts the increase in FGF21, in

association with changes in the microbiome. The conclusion is that in response to cellulose, the microbiome is doing something that mitigates the PR diet. Conversely, Figure 4 demonstrates that the depletion of the microbiome blocks the increase in FGF21. Thus it seems possible that the microbiome is simultaneously necessary for the increase in FGF21 on a PR diet, and the decrease in FGF21 following cellulose. I do not think this is actually a contradiction, but I do feel the manuscript should address these two observations head-on and articulate a plausible mechanism that could explain this biphasic role of the microbiome. Ideally, these ideas could be tested with the available samples.

9. The conclusion of Figure 4 is based on the use of antibiotics to deplete the microbiome. But not data are included which demonstrate this depletion post-antibiotic treatment. The results suggest that suppression was confirmed by qPCR, but this evidence is not provided. I would strongly suggest that some measurements be made to assess and confirm the impact of the antibiotic treatment, as this fact is essential to the outcomes of Figure 4.

10. Prior work suggests that the weight reduction induced by PR diets is mediated by FGF21. If this were true, then one would expect that the cellulose supplementation, by blocking the increase in FGF21, would also block the reduction in body weight. Yet this did not occur. Similarly, one would expect that antibiotic depletion would also block the effect of PR on body weight, but this was not measured. Can the authors provide any insight into this apparent contradiction?

11. Fig 1: mRNA expression graph is hard to read because the y-axis is so large. It looks like PR increased FGF21 by over 10 fold, but is this not significant?

12. Typo in line 198. Should it be dependent?

13. Line 198-201 states: "Cellulose promotes the growth of bacterial species with potential to degrade cellulose in the non-ruminant gut but this does not appear to impact host protein or amino acid status given the fact that antibiotic treatment had no impact on FGF21 levels in the cellulose supplemented groups". I do not understand how this conclusion was reached. The lack of effect in the cellulose groups in Figure 4 is meaningless because there is no effect of PR in the first place and therefore no increase in FGF21 for the cellulose to block. The key observation is that PR has no effect on FGF21. Beyond that the lack of effect of cellulose/insulin has little value. In addition, I would suggest that it is risky to make conclusions on host protein or amino acid status after measuring only FGF21. Instead, I would encourage the authors to actually measure host protein/amino acid status in a comprehensive way, as the question is extremely important for the overall interpretation (related to comments above).

14. A similar conclusion is reached in line Line 230: "However, we concluded that the silencing of the FGF21 stress response occurred independent of the changes in the gut microbiota as demonstrated by the antibiotic intervention." Since PR does not increase FGF21 in the absence of microbiota, it is hard to argue there is no silencing because there is nothing to silence in the first place. One cannot reduce something that is not increased. The point is that the increase in FGF21 seems to require a microbiome, but we have no idea why.

15. Line 239: "We observed that both low and high cellulose supplementation silenced the stress response in a dose-dependent manner." This statement is problematic for 2 reasons. First, there is no PR control so it is not clear that it is the cellulose that is causing the silencing, or instead that inulin is producing a unique effect. The doses of fiber used here are different than the earlier experiment (Figure 1), so one cannot just assume that the effects are the same as Figure 1. Second, the low dose cellulose does not block the increase in FGF21, but does block the increase in ASNS/3PGD and eIF2a phosphorylation. So which of these markers is more appropriate for the 'stress response'.

16. A reduction in body weight without changes in energy intake is consistent with an increase in energy expenditure. FGF21 has been linked to changes in EE. Do the others have any ability to measure changes in energy expenditure?

Reviewer #3:

Remarks to the Author:

The study "Gut Microbiota Mediate the FGF21 Adaptive Stress Response to Chronic Dietary Protein Restriction in Mice" by Anthony Martin and colleagues show dietary protein restriction triggers the FGF21 but supplementing protein restricted diet with prebiotic fiber demonstrate that PR diets supplemented with fibers altered FGF21 but the effect was opposite whether the diet was supplemented by inulin or cellulose. Cellulose decreased the FGF2 response whereas inulin had no effect even though it changed microbial diversity. One of the interesting findings was the enrichment of Parabacteroides spp in the presence of cellulose given that not many bacteria possess capability of utilizing cellulose. Overall this is a well done study with novel findings that link gut bacteria to metabolic responses following dietary changes. Few suggestions are mentioned below-

1. It would be helpful to demonstrate that the fibers are indeed acting via the gut microbiota by including a germ free control experiment where the protein restricted diet is supplemented with the fibers.
2. Parabacteroides spp has previously not been described to utilize cellulose, hence it will be helpful to characterize these strains using whole genome sequencing to help identify potential operon that might be involved in the process.

Dear Reviewers: We apologize that significant time has passed since you last reviewed this manuscript. COVID-19 lab closures forced a delay in our animal experiments, however we were able to ultimately complete them all. We have taken extra care in our responses to explain what has changed from the previous manuscript given that the reviewers may not recall this study in its entirety. All of our responses are annotated in red.

Reviewer #1 (Remarks to the Author):

I think the basic observations here are interesting but don't think it is quite ready for publication. Several aspects of the experimental design are not very clear and it is hard to see how the various factors known to influence FGF21 signaling were experimentally accounted for. As a consequence, the support for the conclusions at present is not sufficiently robust. I don't think there is much to support the claim in the last line of the abstract and the sentence on L38 to L41 is unsupported in two ways – there is no objective way to categorize the microbiota as 'favorable' and there is no meaningful characterization of how it has changed provided.

We thank the reviewer for their interest in our overall finding and agree that the manuscript as presented was not publication-ready. We have substantially revised our study primarily through the addition of several control diet experiments (as recommended by the reviewer as well as Reviewer #2), which now add important comparisons for fully interpreting the data. We have also performed deeper analyses of the target microbiota of interest, and have removed the antibiotic experiment and replaced it with a cleaner germ-free experiment. We feel this manuscript is greatly improved thanks to the constructive reviewer comments.

Regarding the specific comments about the last line of our abstract and the introduction, we apologize if it sounded too presumptuous. We have revised this and intentionally use language such as "the data suggest" and throughout the sentence we say "may" occur, hoping to convey a possible concept rather than proven fact.

Regarding L38-41 in the original text, we agree that favorable is a subjective term and have now removed this in the revision as we have re-written the introduction.

1) The experimental diets need to be more clearly described to understand the rationale and experimental tests. The two basic diets are <10% (specific value should be given) and 18% protein. These are stated to be isocaloric, which presumably means that protein has been replaced with digestible carbohydrates? The experimental test is to examine the impact of fiber supplementation on the outcomes of protein restriction. To this end the two base diets were supplemented with different doses of fiber sources - either inulin or cellulose. Were these fiber-supplemented diets considered isocaloric with the non-supplemented diets? Not so in terms of energy bioavailability. Inulin is readily microbially digestible and so expected to yield ca 2.5 cal/g to the mouse, whereas cellulose is of limited digestibility and typically inferred to be close to 0. If the diets were kept isocaloric as the text implies then other components must also be adjusted - what were these?

We agree and were remiss to have not included that in the original submission. We now have included complete diet formulations (New Supplementary Table 1), and have also included a

schematic of the diet groups (New Figure 1) to make them easier to follow as we have included new diet control groups in this revision.

To answer the specific question pointed out by the reviewer, the macronutrient density of fat (% by weight) was kept constant across all diets, and protein was altered to either create protein-restriction or protein-sufficient status. Therefore, as the reviewer states, the carbohydrates were modified to achieve as closely isocaloric diets as possible. The specific components that were modified to alter the carbohydrate amounts in the respective diets include (1) corn starch, and (2) maltodextrin.

Whenever cellulose or inulin was altered, we adjusted with corn starch because it has the least caloric contribution since it is 10% moisture, (so only 3.6 kcal/g), so this minimized the impact on calories when altering the non-caloric cellulose or reduced calorie inulin (1.2 kcal/g as stated by our diet manufacturer, Envigo). Nonetheless, it is challenging to get kcals exact across nine diets, thus is diet was maintained at ~3.3-3.8 kcal/g as shown in the new table. However, we did aim to keep the diets most closely isocaloric across the primary comparison groups.

2) On L57 - L63 (Fig 1) the supplementations are described as being at 3.5% and 10.5% (presumably by weight). On L97-98 (Fig 2) the supplementations are at 5% and 15%. Clarification of why the change was made should be given.

We understand the confusion and have now revised this. In the original manuscript, we had tested two levels of protein restriction- 10% protein by weight and 8% protein by weight. The reason being that in the second set of experiments we wanted to push the protein level down further to see if the effects we were seeing were dose-dependent, or whether there was simply a critical threshold to be met whereby no further stress signaling could be elicited. Regarding the earlier comment, we had to in-turn alter the level of fiber supplementation to maintain isocaloric levels given that when the protein was reduced we had to increase carbohydrates.

However, because we felt there was significant confusion raised by the other reviewers as well about the different protein levels, in this revision we have only studied the 10% protein diet, and all new experiments were performed with this diet formulation. The only exception is the germ-free experiment in New Figure 5 whereby we fed a fiber-free, 8% protein diet. In our original submission we performed an antibiotic-treated mouse experiment and these mice were on the 8% protein diet. When now performing the germ-free experiment, due to COVID-19 lab closures, these mice were in short supply, therefore rather than switch to the 10% protein diet and introduce a new variable that may require repeating, we chose to keep consistent with the previous experiment.

3) It is surprising that the mice did not increase food consumption in response to protein dilution (Fig. 1b), as has been reported many times previously. This needs to be discussed somewhere.

To further explore the reviewer's point, we added three additional control diets to demonstrate the impact of fiber consumption on the background of protein restriction. These additional control diets also helped us parse out the influence of fiber on food consumption because in the original submission we did not have any fiber-free diets. These new diets are represented in New Figure 1

and are as follows: (1) protein sufficient – fiber-free, (2) protein sufficient + 5% inulin, and (3) protein restricted [10% protein by weight] – fiber-free.

In comparing food consumption between the mice fed a protein sufficient (fiber-free) vs. protein restricted (fiber-free) diet, we now demonstrate that protein restricted mice consumed more kcal/day relative to mice fed a protein sufficient diet. This observation is consistent with previous reports (as noted by the reviewer) and is shown in New Figure 2D.

4) L188 - 196 The change in relative abundance should not be described as a bloom, unless you have evidence that it is due to growth. The growth on M9 with cellulose as sole carbon source is interesting. Was the cellulose added to the medium before autoclaving and if so did you exclude the possibility of growth on sugars released during sterilization? A simple experiment here might be testing the ability of *E. coli* to grow on the same medium.

Thank you for the suggestion and we agree- we have changed the word bloom to “increased in relative abundance”.

Thank you for your insightful suggestions about possible release of carbon through autoclaving. To address this comment, we have now run experiments whereby we added non-autoclaved cellulose to autoclaved M9 minimal media (New Figure 6C). This media was plated prior to inoculation with *P. distasonis* to ensure that the non-autoclaved cellulose did not introduce contaminants. Utilizing non-autoclaved cellulose therefore eliminates the possibility of autoclave-induced release of cellulose sugars.

However, the M9 minimal media itself contains trace amounts of NH_4Cl which could provide a nitrogen growth substrate. To address whether this may also be having a confounding effect on our experiment, we created additional treatment groups whereby we removed NH_4Cl . So, we had M9 +/- cellulose and +/- NH_4Cl . Here we observed attenuated growth across the board when the nitrogen source was removed, that could not be rescued by cellulose supplementation. This suggests that a nitrogen source is of highest priority for *P. distasonis* growth, with further enhancement of growth with a carbon source, specifically cellulose as shown in these new experiments. These data are shown in New Figure 5C.

5) The diet supplementations do not show a clear effect of fiber supplementation in Fig 3. By far the majority of explained variation is due to the transition from baseline to 3 weeks experimental diet. Within the experimental diets there is limited variation but to see this you really need to show a separate ordination at just week 3 to see what is happening.

We agree and thank the reviewer for this comment. The above referenced figure is now New Figure 4A, and has been revised. Upon adding the three additional new control diets, we have reanalyzed the 16S rRNA gene sequencing data to include these groups in our beta-diversity analysis (New Figure 4A). As advised by the reviewer, we now focus our ordination on the three-week endpoint which does indeed reveal clearer separations between the diets.

6) Fig 4c shows rather modest evidence for growth. The viable counts for *Lactobacillus* are exceedingly low. A ten-fold change in cfu/ml could easily be growth on trace carbon sources either

introduced with the inulin or released during autoclaving. Fig 4e is a little more convincing with a 100-fold change but this is still about 4 orders of magnitude less than what would typically expect for a bacterial growth experiment. Ideally this would be a growth curve, and in any event some details on the time of incubation and initial inoculum are needed to interpret.

With the addition of new diets and therefore new 16S rRNA gene sequencing data, we re-analyzed the sequencing data and *P. distasonis* once again emerged as the candidate bacterium that was differentially abundant in our cellulose supplemented groups *in-vivo* (New Supplementary Figure 1A-B). With the addition of the new control groups, this made *P. distasonis* more interesting as a target, so in this revision we chose to focus on this point and performed a more detailed growth curve of *P. distasonis* under the +/- cellulose and +/- NH₄Cl growth conditions as described above (New Figure 5C).

Details on the protocol for the bacterial growth curve assay are now included in the Methods section (lines 382-390). Briefly, *P. distasonis* was resuspended in PBS and normalized to an OD₆₀₀ reading of 0.2. In a 96-well format, 175 ul of each media was added to the plate and inoculated with 25 ul of normalized *P. distasonis*. At the respective time points, 50 ul of the inoculated media was plated onto chocolate blood agar plates and incubated for 48 hours prior to quantification of CFUs.

Reviewer #2 (Remarks to the Author):

This manuscript by Martin and colleagues describes the interesting observation that the induction of FGF21 in response to dietary protein restriction is attenuated when the PR diet is supplemented with cellulose but not inulin, and that these changes in FGF21 secretion are associated with unique changes in the gut microbiota. In addition, the ability of PR to increase FGF21, and thus cellulose to reduce it, are lost in mice with antibiotic based depletion of the gut microbiome. Collectively the data lead to the interesting conclusion that the response to a protein restricting diet, at least in terms of FGF21 production, somehow depends on an interaction between protein content, dietary fiber, and gut microbiota.

The strength of the manuscript is clearly the novelty of this interaction between FGF21, fiber and the gut microbiome. While a number of studies have clearly demonstrated that protein restriction increases FGF21, this link to the microbiome appears to be unique.

The primary weakness of the manuscript is that the work describes a phenomenon that is largely unexplained. We have little insight into the mechanism through which cellulose lowers FGF21 but inulin does not, the mechanism through which antibiotic based depletion of the microbiome blocks the effect of PR on FGF21, or the larger implications of this work to metabolic responses to PR or dietary fiber. There are also concerns related to design and interpretation. These issues are discussed in more detail in the specific comments below

We thank the reviewer for their positive comments and agree with the noted weaknesses in the original submission. We have now added three new control diets and have re-analyzed the data with these new diets that we feel addresses a number of the reviewer's concerns and has resulted

in a meaningfully improved manuscript. These revisions are described in detail below.

Specific Comments

1. The diet compositions are not completely clear. The methods reference Table S1 for dietary composition, but Table S1 is the primer sequences. Thus the manuscript does not actually describe the diets in detail, and this leads to confusion that permeates the remaining comments.

We apologize for this oversight. Our diet tables were accidentally dropped from the original version. We have now included these complete diet formulations in New Supplementary Table 1.

2. It is not clear what the appropriate control is for these studies, and whether some of the experiments (Figs 2 and 3) include an appropriate control. There are several related issues:

a. The studies in Figs 2 or 3 lack a PR alone group, yet in Fig 1 and 4 there is a PR alone group. This inconsistency is awkward and raises the concern that there is a missing control (PR alone) in Figs 2 and 3. A key conclusion is that the addition of cellulose blocks the effect of PR. Yet without a PR alone group there is no way to compare the effect of added cellulose or inulin. For instance, in Fig 2 there is no direct evidence that PR increases eIF2a. Second, the result section says eIF2a phosphorylation was ‘suppressed by cellulose’, but it is impossible to make that conclusion without a PR alone group because there is nothing for it to be suppressed relative to. One can only say it is similar to control. It is also not clear if inulin is producing unique effects, for instance exerting an independent effect to increase FGF21, or if it is simply not having any effect and thus FGF21 is increased solely due to PR. Finally, it noteworthy that none of the experiments test whether cellulose or inulin exert effects in the absence of PR. I would not expect them to, but the current data never address that possibility or include these group control+cellulose or control+inulin groups. Although unlikely, it seems possible that inulin supplementation to a control diet alone might be sufficient to increase FGF21, and if so this would greatly cloud the interpretation.

We understand the points the reviewer makes about the need for additional controls, and agree that including additional protein-restricted and protein-sufficient groups without fiber would allow for more robust data interpretation. Therefore, we have added three additional control diets to address these concerns and are now diagrammed in New Figure 1. The new diets include: (1) protein sufficient – fiber-free, (2) protein sufficient + 5% inulin (to compare to the pre-existing protein restricted + 5% inulin group), and (3) protein restricted– fiber-free. In doing so, we have re-analyzed our data to include these new diets and provide better clarity. Furthermore, because of significant confusion raised by the reviewers in the original manuscript regarding different protein levels used, in this revision we have only focused on the 10% protein restricted diets. The only exception is the germ-free experiment in New Figure 5 whereby we fed a fiber-free, 8% protein diet. In our original submission we performed an antibiotic-treated mouse experiment and these mice were on the 8% protein diet. When now performing the germ-free experiment, due to COVID-19 lab closures, these mice were in short supply, therefore rather than switch to the 10%

protein diet and introduce a new variable that may require repeating, we chose to keep consistent with the previous experiment.

We are now able to comprehensively determine the differential effect fiber has on both a protein sufficient and/or protein restricted diet.

b. More generally, it is not clear what the appropriate controls should be. The Envigo AIN-93G diet, which I think but am not certain (see point #1) is the control, contains 5% cellulose. Thus the control and PR alone diet (which is fiber-free) differ not just in protein content, but also in cellulose content. Is it possible that the increase in FGF21 with the PR alone diet (Fig 1) is partly due to the reduction in cellulose? Should the control diet (or a control diet) also have zero cellulose like the PR diet? Conversely, one of the diets is PR plus 5% cellulose. If the control has 5% cellulose, then one could argue that this diet is the true PR alone group and not the fiber-free PR diet. Particularly when one considers that the PR diet and the PR+3.5% cellulose diet in Fig 1 do not produce the same effects. The manuscript at least needs to be clearer about the fiber content in the control (possibly labeling it), and more directly deal with these issues of variance in fiber content and appropriate control. Ideally, if a 5% fiber control is used then the PR alone diet should also be 5% fiber and all comparisons should be added fiber beyond this number. Alternatively, a fiber-free control and fiber-free PR group used, and then cellulose or inulin added to these diets. It is also possible that I am mistaken regarding the actual control since it is not directly described, and if so then many of my subsequent comments might be off base. Regardless, this issue needs to be clarified.

To clarify, conventional rodent diets are supplemented with 5% cellulose. So, our protein sufficient diet + 5% cellulose, is actually standard chow (but we used a defined diet rather than chow to control for variations in carbohydrate sources). This is the typical control diet in most animal studies, however since we were actually testing cellulose levels as a treatment group, we also used this as a comparison to 15% cellulose. This is where a lot of the reviewer's confusion understandably arises.

Nonetheless, to eliminate these confusions, the additional control diets were added (mentioned in response to the previous question 2a). The appropriate diet controls for this study are now noted as: (1) protein sufficient- fiber-free, (2) protein sufficient + 5% inulin (to compare to the pre-existing protein restricted + 5% inulin), and (3) protein restricted- fiber-free. Thus, we can compare the differential effect of protein restriction or repletion in the presence or absence of fibers. We have taken extra care to clarify this in the figures and manuscript main text.

c. There is inconsistency in the dietary fiber content and endpoints measured. Figure 1 compares the effects of a 0%, 3.5% and 10.5% fiber, while figure 2 compares only 5% and 15%. It is not clear that 3.5% and 5% fiber are the same, or that 10.5 and 15% is the same. Figure 1 measures physiological endpoints, while Figure 2 does not. Figure 2 measures liver molecular endpoints, which Fig 1 does not. It is difficult to get a complete picture of what is happening due to the inconsistencies in diets used. It not clear why all the experiments did not use the same diets so that direct comparison between the various figures could be more easily made.

We understand the confusion, which was also raised by Reviewer #1, and have now revised this. In the original manuscript, we had tested two levels of protein restriction- 10% and 8% protein by weight. The reason being that in the second set of experiments we wanted to push the protein level down further to see if the effects we were seeing were dose-dependent, or whether there was simply a critical threshold to be met whereby no further stress signaling could be elicited. We had to in-turn alter the level of fiber supplementation to maintain isocaloric levels given that when the protein was reduced we had to increase carbohydrates. So while the ratios of fiber to protein were kept constant across the groups, the percentages had to change, thus creating confusion.

We did not ultimately observe much difference between 8% and 10% protein, therefore in this revision we have only studied the 10% protein diet. And all new experiments testing fiber supplementation use either 5% or 15% supplementation.

d. In summary, the point of these comments is that there are multiple moving parts among these diets and inconsistency between experiments in the diets used. The diets need to be designed so that either protein or fiber is manipulated/compared, but not both at the same time. The diets need to appropriately consider and articulate the fiber content of the controls, and appropriate controls included in all studies.

We greatly appreciate these constructive comments and have applied them to build a more compelling story that we have re-organized to improve reader clarity.

3. Does PR itself alter the microbiota? Without a PR alone group, it is impossible to address this question, and therefore it is unclear if the addition of cellulose or inulin is reversing an effect of PR on the microbiota, or instead doing something unique to the microbiota that then exerts downstream effects on FGF21.

The addition of the fiber-free, PR group is now reflected in our 16S rRNA gene sequencing analysis to help answer the reviewer's question about the role microbiota play in altering the FGF21 response. In comparing PS and PR (both in the absence of fiber), we now see that these groups cluster separately from each other in our beta-diversity analysis (New Figure 4A).

4. In Figure 2 there is some discrepancy between changes in FGF21 and changes in *ASNS*/*3pgd*, specifically for the 5% cellulose PR group. These latter genes are markers of the liver response to protein restriction, yet neither go up with either cellulose group. As noted above, there is some concern that the Control group also has 5% cellulose, and therefore one wonders if the 5% cellulose group is the actual PR control. But if that were true there should have been an increase in 5% cellulose group. Thus one wonders if the increase in *ASNS*/*3PDG* (which are not measured in Fig 1 but should have been), are being driven by the addition of inulin, and not by the exposure to PR? As I see it, there is no way the figure can argue against this conclusion without a PR alone group.

We absolutely agree and with the addition of the new control groups we can now better evaluate this response. (Note: Figure 2 referenced in the comment is now Figure 3). These data now show that inulin supplementation results in elevated *Asns* and *3pgd* (but an order of magnitude less in *3pdg*) gene expression (New Figure 3C-D). The fiber-free PR group, compared to fiber-free PS

group, also shows a significant increase in expression in both *Asns* and *3pgd*, and this coincides with the increase in FGF21 shown for this group (Figure 2b). Thus protein restriction alone can indeed stimulate the liver amino acid synthesis response, and inulin is unable to mitigate this however cellulose can. We can now more confidently draw these conclusions with the addition of fiber-free control groups.

5. I am not sure I agree with the initial discussion of Figure 3. This discussion focuses primarily on the inulin groups, and indeed the title of this section (line 116) references inulin. However, when I focus on Figure 3A the primary observation is the unique clustering of the 15% cellulose group. This group seems more divergent from control than inulin, or at least similarly divergent (but in a different direction). Considering this it is this 15% cellulose group that exerts the unique effect of blocking PR, why isn't this group the primary emphasis? Perhaps I am missing something.

(Note: Figure 3A referenced above is now Figure 4A)

No, the reviewer is not missing something. We were perhaps concerned about over-emphasizing the cellulose-microbiome relationship because it is so engrained in our microbiome thinking that cellulose is almost inert. So rather than set off a wave of skepticism, we initially chose to frame our results around inulin. However, now with the addition of the control groups, we feel quite strongly about our conclusions around cellulose, which the reviewer notes is striking. We have shifted this section now to focus on the cellulose aspect. Upon the addition of our three new diet control groups we have also re-analyzed this data which gave us an opportunity to appropriately re-focus our message. Indeed, the PR+15% cellulose still emerges as the most divergent group as represented in our beta-diversity analysis in New Figure 4A. We have now modified New Figure 4 and our manuscript text to highlight the divergence of this group.

6. Fig 4: The only in vivo physiological endpoint measured in Figure 4 is FGF21. The manuscript would be much stronger if additional endpoints related to hepatic amino acid sensing (eIF2a, ASNS, etc) were measured, as well as whole animal measures of the larger metabolic response to PR. Currently there are several important unanswered questions: 1) Is the effect of microbiome depletion specific to FGF21, or a more general effect on the liver sensing of PR? 2) Does the depletion of the microbiome also block changes in body weight, etc that occur in response to PR, and are these related to changes in FGF21?. It seems very insufficient to just measure FGF21. More generally, it would be nice to simply see the metabolic/bodyweight state of the depleted mice, to get some insight into their health.

(Note: Figure 4 referenced above is now New Figure 5).

We agree with the reviewer, and have actually had the opportunity to run this experiment in germ-free mice which we feel is a much cleaner model than antibiotic depletion for answering this question. As a result we were able to run the analyses mentioned by the reviewer.

To recap, in the previous version of the manuscript we conducted a microbiota depletion experiment using antibiotics in conventional C57Bl/6 mice fed a protein sufficient or protein restricted diet. This study showed that with a minimal microbiota, mice could not mount an attenuated FGF21 response to protein restriction. This indicated that gut microbiota are necessary for the FGF21 response. Unfortunately, we did not have sufficient material to perform the

additional hepatic amino acid sensing assays as the reviewer noted above. Thus, in the previous manuscript version we only presented our analysis of FGF21.

In this revision, we have instead performed a germ-free mouse experiment under protein restriction (fiber-free, PR) which we believe serves as a more robust model. Indeed, the data confirm that the absence of microbiota result in blunted FGF21 signaling, more completely so than with antibiotic treatment. In addition to plasma and liver FGF21 levels, we now show hepatic gene expression for *Asns* and *3pgd* as requested by the reviewer, and these data reveal that these liver amino acid synthesis genes are also blunted in the germ-free mouse. Furthermore, we also include body weight data to compare how the presence or absence of gut microbiota impact the overall growth trajectory of the mouse in response to protein restriction. The comparison in this new figure is to conventional mice fed the fiber-free, PR diet. Most interesting is that our germ-free liver and growth data look very similar to results in FGF21 KO mice (Laeger, T., et al. *Cell reports* **16(3)**, 707–716. (2016).).

7. A major concern of the manuscript is that the effects on FGF21 are largely phenomenological. We have very little insight into what the cellulose is specifically doing that would rescue liver FGF21 production. We also have very little insight into the mechanism through which the depletion of the microbiome blocks the increase in FGF21 during PR. Are these even the same mechanism? Some of these questions might be answered by more thoroughly testing and phenotyping the studies already conducted as noted above. There are several obvious possibilities, but one very important one which I think should be addressed in this manuscript is the possibility that the microbiome is synthesizing amino acids, which are absorbed and detected by the liver, thus preventing the liver response to protein restriction (increased FGF21, ASNS, 3PDG). This question would be partially tested by measuring ASNS/3PDG in the mice in Figure 1 and Figure 4, and as noted above by including appropriate controls in Figure 2. In addition, it would seem possible, but more difficult, to actually measure microbial amino acid production and/or absorption. In other words, is the addition of cellulose preventing the liver from sensing protein restriction (ASNS not increasing, for instance). Or does the liver sense the PR normally, but there is something unique occurring to FGF21?

We completely agree with the reviewer that these are outstanding and important questions, but feel at this stage these are beyond the scope of the existing manuscript. Measuring microbial amino acid production *in vivo*, accurately, is notoriously challenging as true free amino acid levels in the blood are a calculation of rate-of-appearance minus rate of degradation, and it is challenging to measure degradation reliably. We also are uncertain whether it is free amino acids that are the signaling molecule. We have investigated what is required to begin to answer these questions, and these involve stable isotope tracer studies of labelled ammonia in both the diet and in separate experiments, injected. As well as *in vitro* stable isotope experiments in the bacterial culture media. However, it may not be amino acids at all that are the signal and may be a small peptide or other microbial metabolite. For these we would take an untargeted mass spec approach of blood and tissues under protein restriction. We plan to perform these in both conventional and gnotobiotic mice. These are complex experiments that we feel will be a paper unto itself. We feel that with the addition of the new experiments to this current manuscript we have laid a strong foundation and case for pursuing this line of investigation in a follow-up paper with a focus on targeting the signaling molecule.

This being said, we do now explain in further detail, in newly generated whole genome sequencing (WGS) data, that *P. distasonis* possesses genes specifically related to amino acid production and essential amino acids in particular. These amino acid biosynthetic genes are now listed in New Table 1. Furthermore, our new *in vitro* experiments also show that *P. distasonis* maintains growth in minimal media supplemented with cellulose but not with inulin. Interestingly, when we exclude a nitrogen source (NH₄Cl) from the minimal media the growth of *P. distasonis* is attenuated. This demonstrates that cellulose supplementation alone is not sufficient to maintain growth over time. This further suggests that a nitrogen source is of highest priority for *P. distasonis* growth, with further enhancement of growth with a carbon source (cellulose). These data are shown in New Figure 6C. The combination of the new WGS and *in vitro* experiments provide a plausible mechanism by which *P. distasonis* may utilize nitrogen and fiber (carbon source) in combination to produce amino acids possibly for host uptake.

8. There is a disconnect of sorts between the data in Figure 1-3 and that of Figure 4. The first several experiments suggest that the addition of cellulose blunts the increase in FGF21, in association with changes in the microbiome. The conclusion is that in response to cellulose, the microbiome is doing something that mitigates the PR diet. Conversely, Figure 4 demonstrates that the depletion of the microbiome blocks the increase in FGF21. Thus it seems possible that the microbiome is simultaneously necessary for the increase in FGF21 on a PR diet, and the decrease in FGF21 following cellulose. I do not think this is actually a contradiction, but I do feel the manuscript should address these two observations head-on and articulate a plausible mechanism that could explain this biphasic role of the microbiome. Ideally, these ideas could be tested with the available samples.

We agree with the reviewer in that our data suggests that there may indeed be a dichotomous role that microbiota play in the FGF21 response.

We now state in the Discussion on lines 286-293:

“We also discovered that FGF21 is non-responsive in protein restricted GF mice, demonstrating a requirement for the gut microbiome in mediating this nutritional stress response. When the gut microbiota are present, FGF21 signals appropriately in response to protein-restriction, as demonstrated in conventionally-raised mice. So in the absence of a gut microbiota, the FGF21 response to a PR diet is lost and animals grow normally despite decreased protein. However, we find this can also be achieved with an intact microbiome if cellulose is supplemented. This suggests there is a microbially produced signaling molecule, whether it is free amino acid production or other microbial metabolite, that is mitigating FGF21.”

9. The conclusion of Figure 4 is based on the use of antibiotics to deplete the microbiome. But not data are included which demonstrate this depletion post-antibiotic treatment. The results suggest that suppression was confirmed by qPCR, but this evidence is not provided. I would strongly suggest that some measurements be made to assess and confirm the impact of the antibiotic treatment, as this fact is essential to the outcomes of Figure 4.

In the revised manuscript, we have now performed a germ-free experiment (New Figure 5) as we believe it is a more robust model, and no longer include the antibiotic-treated group.

10. Prior work suggests that the weight reduction induced by PR diets is mediated by FGF21. If this were true, then one would expect that the cellulose supplementation, by blocking the increase in FGF21, would also block the reduction in body weight. Yet this did not occur. Similarly, one would expect that antibiotic depletion would also block the effect of PR on body weight, but this was not measured. Can the authors provide any insight into this apparent contradiction?

With the addition of our new diet controls and the germ-free experiments, we now introduce new body weight trajectories for both our conventional and germ-free experiments (New Figure 2C and 5e). In our revised manuscript, we observe that the supplementation of 5% cellulose or 15% cellulose on the background of a PR diet mitigates the FGF21 response to dietary PR as was previously shown (New Figure 2A-B). The observed reduction of FGF21 signaling upon cellulose supplementation corresponds with the attenuation of weight loss in both cellulose supplemented groups, and grow comparably to the fiber-free, protein sufficient control group (New Figure 2C). Our protein restricted germ-free study further demonstrates that in the absence of gut microbiota, FGF21 signaling is lost (New Figure 5A-B), and we indeed observe in these mice, as the reviewer points out, that there is no reduction in body weight gain and actually demonstrates a greater body weight gain than the conventional mouse (New Figure 5E). As we note, this data is consistent with studies of FGF21 KO mice.

11. Fig 1: mRNA expression graph is hard to read because the y-axis is so large. It looks like PR increased FGF21 by over 10-fold, but is this not significant?

(Note: Figure 1 referenced above is now Figure 2)

Thank you for calling this to our attention, the 10-fold increase in *Fgf21* expression was in fact significant and the lack of statistical representation in the original figure was a typo. In our revised manuscript where we focus our analyses on PR at 10% protein by weight, we observe a significant increase in *Fgf21* expression in response to a fiber-free, PR diet relative to the fiber-free PS group (New Figure 2B).

12. Typo in line 198. Should it be dependent?

This is indeed a typo, thank you for calling this to our attention, we have now corrected it.

13. Line 198-201 states: “Cellulose promotes the growth of bacterial species with potential to degrade cellulose in the non-ruminant gut but this does not appear to impact host protein or amino acid status given the fact that antibiotic treatment had no impact on FGF21 levels in the cellulose supplemented groups”. I do not understand how this conclusion was reached. The lack of effect in the cellulose groups in Figure 4 is meaningless because there is no effect of PR in the first place and therefore no increase in FGF21 for the cellulose to block. The key observation is that PR has no effect on FGF21. Beyond that the lack of effect of cellulose/insulin has little value. In addition, I would suggest that it is risky to make conclusions on host protein or amino acid status after measuring only FGF21. Instead, I would encourage the authors to actually measure host protein/amino acid status in a comprehensive way, as the question is extremely important for the overall interpretation (related to comments above).

Thank you for this comment, and we agree with the comment that fiber supplementation would have no effect on PR responses in the absence of gut microbiota. We have significantly overhauled these experiments through the addition of the new control groups and the addition of the germ-free experiments such that we feel much of the reviewer's concern is addressed. However, in light of the above comment, when we conducted the germ-free experiments we only compared protein-restricted vs protein-sufficient and did not perform a fiber supplementation, for the reasons the reviewer noted. We also agree with the reviewer's second comment and meant only to speculate on this idea as a hypothesis, rather than a conclusion from the data. We have tempered our language accordingly and make no conclusions about the amino acid status of the host since we do not measure this directly for the reasons noted in response to question #7.

14. A similar conclusion is reached in line Line 230: "However, we concluded that the silencing of the FGF21 stress response occurred independent of the changes in the gut microbiota as demonstrated by the antibiotic intervention." Since PR does not increase FGF21 in the absence of microbiota, it is hard to argue there is no silencing because there is nothing to silence in the first place. One cannot reduce something that is not increased. The point is that the increase in FGF21 seems to require a microbiome, but we have no idea why.

We agree and have removed this comment as part of the revised section on the germ-free mouse experiments.

15. Line 239: "We observed that both low and high cellulose supplementation silenced the stress response in a dose-dependent manner." This statement is problematic for 2 reasons. First, there is no PR control so it is not clear that it is the cellulose that is causing the silencing, or instead that inulin is producing a unique effect. The doses of fiber used here are different than the earlier experiment (Figure 1), so one cannot just assume that the effects are the same as Figure 1. Second, the low dose cellulose does not block the increase in FGF21, but does block the increase in ASNS/3PGD and eIF2a phosphorylation. So which of these markers is more appropriate for the 'stress response'.

Thank you for these comments. Many of these concerns have been addressed experimentally through the addition of the new control diets.

Specifically, in regard to the second comment, upon addition of the new fiber-free, PR control group we can now compare the effect of fiber more clearly. Here we observe that the addition of 15% cellulose mitigates phospho-eIF2a, FGF21, *Fgf21*, *Asns*, and *3pgd* gene expression relative to the fiber-free, PR group (New Figure 2 and 3). The PR+5% cellulose group performed similarly on all metrics except FGF21 plasma level as it was not significantly different relative to the fiber-free, PR group (New Figure 2A). However, 5% cellulose trended in the same direction with reduced circulating levels of FGF21 and was not statistically different from the fiber-free protein sufficient (New Figure 2A).

16. A reduction in body weight without changes in energy intake is consistent with an increase in energy expenditure. FGF21 has been linked to changes in EE. Do the others have any ability to measure changes in energy expenditure?

We agree this is an interesting question and note that increased energy expenditure is observed in mice with reduced FGF21 levels. Unfortunately we do not have the capacity to measure energy expenditure, and apologize as this would be interesting. We do recognize the utility of the measurement and have noted both the literature showing this relationship (Lines 28-29) and the acknowledgement that we could measure it in this study (Lines 88-90).

Reviewer #3 (Remarks to the Author):

The study “Gut Microbiota Mediate the FGF21 Adaptive Stress Response to Chronic Dietary Protein Restriction in Mice” by Anthony Martin and colleagues show dietary protein restriction triggers the FGF21 but supplementing protein restricted diet with prebiotic fiber demonstrate that PR diets supplemented with fibers altered FGF21 but the effect was opposite whether the diet was supplemented by inulin or cellulose. Cellulose decreased the FGF2 response whereas inulin had no effect even though it changed microbial diversity. One of the interesting findings was the enrichment of *Parabacteroides* spp in the presence of cellulose given that not many bacteria possess capability of utilizing cellulose. Overall this is a well-done study with novel findings that link gut bacteria to metabolic responses following dietary changes. Few suggestions are mentioned below-

We thank the reviewer for their support of our study and address your specific comments below.

1. It would be helpful to demonstrate that the fibers are indeed acting via the gut microbiota by including a germ-free control experiment where the protein restricted diet is supplemented with the fibers.

Thank you for this comment and we agreed that a germ-free experiment would be much more robust than the antibiotic treatment we previously showed. We have now performed a germ-free experiment to determine the role that gut microbiota play in the FGF21 stress response to protein restriction (New Figure 5). We discovered that in the absence of a gut microbiota the FGF21 stress response to protein restriction is completely blunted. We did not include a fiber supplemented group however, because if FGF21 is already blunted, the addition of fiber (particularly cellulose) would not elicit any response since there would be no response to mitigate.

2. *Parabacteroides* spp has previously not been described to utilize cellulose, hence it will be helpful to characterize these strains using whole genome sequencing to help identify potential operon that might be involved in the process.

Thank you for this excellent comment. We have now performed WGS on our *P. distasonis* strain and did not find any specific operons that would provide insight on its cellulolytic capacity specifically. However, several enzymes did come up that were associated with carbohydrate fermentation that are responsible for breaking carbon linkages very similar to those found in cellulose. For instance, within the functional genomic categories mapping to carbohydrate metabolism we observed the presence of hydrolytic enzymes responsible for breaking down oligosaccharides including several glucosidases (i.e. oligo-1,6-glucosidase & alpha-glucosidase) and xylanase. We now show these WGS data in New Figure 6D, New Supplementary Figure 2, and Supplementary Table 2. Furthermore, we performed more in-depth growth curve assays with *P. distasonis* testing +/- a nitrogen source in addition to +/- inulin and +/- cellulose. This is shown in New Figure 6C.

Reviewers' Comments:

Reviewer #1:

Remarks to the Author:

The authors have done a terrific job of addressing the various questions raised. I have one remaining query, regarding the evidence for growth of *P. distasonis* on cellulose (or inulin).

Fig 6 basically shows a 6 fold increase in colonies over 24 hours - that is less than three cell divisions. The growth curve for no carbon source (only added nitrogen) is the same at 12 hours as the cellulose and inulin. It is only at 24 hours that there is a difference - and that is due to loss of viable cells not ongoing growth. Viable count at 12 hours on the three media containing nitrogen is distinct from the two media with no nitrogen - what growth has occurred to that time is irrespective of presence/absence of a carbon source. Viable count at 24 hours on cellulose is higher than on inulin or no added carbon source BUT has not increased from 12 hours. The difference cannot be attributed to growth on cellulose? Unless I am missing something, this needs some comment.

Reviewer #2:

Remarks to the Author:

The authors have been very responsive, and the resulting revision is much stronger than the original. There is more clarity in the experiments and outcomes, and new experiments and data were included.

Strengths.

1. The core observation is novel. I agree that these are the first data indicating that fiber and/or the microbiome influences FGF21 production.
2. The overall conclusion is provocative, again suggesting that there is an interaction between dietary protein restriction, the microbiome, and FGF21.
3. The manuscript is well written, and the data are presented clearly.
4. The analysis of the microbiome appears well-considered, although I am not an expert in this area.

Weaknesses

1. Despite the novelty of the core observation, the manuscript provides little insight into the underlying mechanism and as such is largely phenomenological. Mice were placed on a variety of diets and a variety of endpoints were measured. This is not to take away from the work, but for publication in a journal such as Nature Communications, usually some specific insight into the underlying mechanism is expected. This lack of any meaningful understanding or explanation for why the effects are observed also leads to other issues, as noted below.
2. Somewhat related to the above comment, what is actually happening in these experiments remains unclear. For instance, it does not completely make sense that that FGF21 induction in response to PR would be both dependent on the microbiome and suppressed by cellulose via the microbiome. In the discussion, the authors suggest that: "...there is a microbially produced signaling molecule, whether it is free amino acid production or other microbial metabolite, that is mitigating FGF21." I do not agree with this conclusion, because this signal cannot be there if there are no bacteria, yet the effect of PR is lost in GF mice. Instead, the data suggest that the microbiome is required for FGF21 induction (not mitigation). This then begs the question of what is happening with cellulose, with one possibility being that cellulose disrupts this normal, microbiome-dependent signal. But of course, this is all speculation because there is no insight into the underlying mechanism.
3. The manuscript never demonstrates that the effect of cellulose is microbiome dependent, but only assumes this relationship. As such, the effect of cellulose and the lack of effect in GF mice may be mediated by completely different mechanisms and thus not connected.
5. Although the manuscript does a much better job of providing clear controls, it still is missing a

PS +15% Cellulose group. While not necessarily relevant for FGF21, it would be useful in other ways.

6. The inulin group is particularly unusual and it is not clear how valuable this group is for comparisons. First, 5% inulin alone suppresses growth equivalently to PR, and there is no 15% inulin alone group. Second, inulin alone also increased liver FGF21, increased p-eIF2a, and increased ASNS. Thus inulin alone was sufficient to trigger a PR-like response in the liver. As such, interpretation of the PR+inulin groups is problematic, because it is difficult to know what is inulin and what is PR. The result is that inulin does not offer any value as a comparator for cellulose, and the manuscript lacks any clear explanation for why inulin is producing these divergent effects

7. Interpretation of the food intake data in Fig 1d is problematic. 15% cellulose appears to completely block the PR-induction of FGF21, yet it fails to block the increase in food intake. Conversely, inulin supplementation fails to block the increase in serum FGF21, but at all doses blocks the increase in food intake. These observations thus contradict recent evidence suggesting that FGF21 drives PR-induced hyperphagia, although it is certainly possible that FGF21 is not the primary driver of PR-induced hyperphagia.

8. The study in germ-free mice is nice and provocative, but also incomplete. First, the experimental only compares conventional and GF mice on PR, and thus lacks dietary controls. Ideally, the experiment should have been a 2x2 with both diet and germ-free status as effects (4 total treatments). The lack of controls is important because it is hard to define the extent to which the GF mice fail to respond to PR without a control for comparison. As such, it is not clear if the response to PR is completely lost or partially blunted. It also seems possible that GF mice could have baseline differences in important endpoints even in the absence of PR, although I acknowledge that I am less concerned for the molecular endpoints (FGF21, ASNS, etc), which would generally be low in mice on a control diet. Finally, although the failure of GF to respond to PR is an interesting observation, it must be acknowledged that the manuscript lacks any clear explanation for why this failure occurs.

Reviewer #3:

Remarks to the Author:

All comments have been addressed appropriately.

RESPONSE TO REVIEWER COMMENTS

Reviewer #1 (Remarks to the Author):

The authors have done a terrific job of addressing the various questions raised. I have one remaining query, regarding the evidence for growth of *P. distasonis* on cellulose (or inulin).

Fig 6 basically shows a 6-fold increase in colonies over 24 hours - that is less than three cell divisions. The growth curve for no carbon source (only added nitrogen) is the same at 12 hours as the cellulose and inulin. It is only at 24 hours that there is a difference - and that is due to loss of viable cells not ongoing growth. Viable count at 12 hours on the three media containing nitrogen is distinct from the two media with no nitrogen - what growth has occurred to that time is irrespective of presence/absence of a carbon source. Viable count at 24 hours on cellulose is higher than on inulin or no added carbon source BUT has not increased from 12 hours. The difference cannot be attributed to growth on cellulose? Unless I am missing something, this needs some comment.

We thank the reviewer for their support of the revised manuscript and for their insightful comment. The reviewer has made a discerning observation and highlights how our language could be improved to more accurately describe the heightened performance of the cellulose supplement with regard to our *P. distasonis* cultures. As the reviewer notes, all of the +N groups had equal growth at 12 hrs, and it was only after this point that significant differences emerged. Notably, these differences were not due to continued cell division and growth in the cellulose group, but rather that *P. distasonis* began to die off in the other groups, while the cellulose helped to maintain viability, which we acknowledge is not the same as continued growth. We do not feel this affects the overall interpretation that cellulose outperforms inulin or lack of fiber in supporting this organism, but we have revised our description of this data to better articulate our findings and now incorporate the reviewer's comments (Lines 246-250) and copied below:

"Additionally, we tested whether the presence or absence of nitrogen in the media would affect the growth of this isolate to determine the hierarchy of substrate requirements for P. distasonis. For the first twelve hours of the time course, P. distasonis grew comparably in minimal media alone or with cellulose or inulin supplement (Fig. 6c; solid lines). However, viable P. distasonis counts dramatically decreased in both minimal media alone and with inulin supplement while cellulose supplementation sustained survival of the bacterium through the end of the time course."

Reviewer #2 (Remarks to the Author):

The authors have been very responsive, and the resulting revision is much stronger than the original. There is more clarity in the experiments and outcomes, and new experiments and data were included.

Strengths.

1. The core observation is novel. I agree that these are the first data indicating that fiber and/or the microbiome influences FGF21 production.
2. The overall conclusion is provocative, again suggesting that there is an interaction between dietary protein restriction, the microbiome, and FGF21.
3. The manuscript is well written, and the data are presented clearly.
4. The analysis of the microbiome appears well-considered, although I am not an expert in this area.

We thank the reviewer for their positive comments regarding our additional efforts in this revision.

Weaknesses

1. Despite the novelty of the core observation, the manuscript provides little insight into the underlying mechanism and as such is largely phenomenological. Mice were placed on a variety of diets and a variety of endpoints were measured. This is not to take away from the work, but for publication in a journal such as Nature Communications, usually some specific insight into the underlying mechanism is expected. This lack of any meaningful understanding or explanation for why the effects are observed also leads to other issues, as noted below.

We acknowledge that a clear mechanism for the mitigation of the FGF21 response despite protein restriction is not presented in this manuscript. However, we do feel we have parsed out deeper insight on the microbial aspects of this study with regard to interactions with cellulose vs. inulin, which is also central to the study. The reviewer acknowledges this in the previous review with a similar comment regarding phenomenology and mechanism stating, with regard to the microbial mitigating effect on FGF21,; *“...There are several obvious possibilities, but one very important one which I think should be addressed in this manuscript is the possibility that the microbiome is synthesizing amino acids, which are absorbed and detected by the liver, thus preventing the liver response to protein restriction (increased FGF21, ASNS, 3PDG). This question would be partially tested by measuring ASNS/3PDG in the mice in Figure 1 and Figure 4, and as noted above by including appropriate controls in Figure 2. In addition, it would seem possible, but more difficult, to actually measure microbial amino acid production and/or absorption. In other words, is the addition of cellulose preventing the liver from sensing protein restriction (ASNS not increasing, for instance). Or does the liver sense the PR normally, but there is something unique occurring to FGF21?”*.

We absolutely agree with the reviewer on this and our primary hypothesis is indeed that there is microbial amino acid production which is contributing to host amino acid status in the dietary protein restricted state. However, there are no currently available tools to measure this accurately nor directly. We are actually working on tool-development for this specific question, testing different stable isotope approaches, working with experts in isotope labeling and amino acid metabolism, but there are multiple physiological and technical challenges we are working through that require further deconstructing of the approach, and a continuous iterative set of experiments to get closer to a method that works. We feel we will get there, but not in a timeframe that can be of utility for this manuscript. However, we took the suggestion made by the reviewer for partially answering this, by including the additional dietary controls and measuring ASNS and 3PGD in the livers of the mice, which proved informative as the reviewer suspected.

These data, plus the addition of new control diets which were protein-sufficient or protein-restricted totally devoid of fibers, highlights that that when protein is sufficient and therefore FGF21 and downstream markers are low, anything else you add to the diet (i.e. fiber) doesn't make any difference to the FGF21 response. When protein is restricted, however, as with the PR + 0% fiber, FGF21, ASNS and 3PGD are elevated, and the addition of cellulose at any level reduces these readouts. This demonstrates that when protein is sufficiently low, then adding a supplement, in this case cellulose, does have an impact. Then with the antibiotic experiments (explained further in response to comment #2), when the microbiota are knocked down, the effects of PR and cellulose are lost. Then we show that the addition of cellulose during PR is promoting the growth of certain bacteria in the conventional mice. In the previous revision, to further address the reviewer's comment above, we performed whole genome sequencing of the primarily enriched bacterium in the PR + cellulose diets, and show that this bacterium

has the genetic machinery to both degrade complex polysaccharides, and produce an array of amino acids including the essential amino acids. Not only that, pathways for carbohydrate metabolism, amino acids & derivatives, and protein metabolism make up nearly half of this bacterium's genome. And we further demonstrate in vitro, through isolation of the organism enriched in the PR + cellulose diets, that this organism's growth is most effectively supported by cellulose. The only missing piece is that we cannot as of yet measure directly that this bacterium is producing amino acids utilized by the host, which is a limitation of currently available technology.

2. Somewhat related to the above comment, what is actually happening in these experiments remains unclear. For instance, it does not completely make sense that that FGF21 induction in response to PR would be both dependent on the microbiome and suppressed by cellulose via the microbiome. In the discussion, the authors suggest that: "...there is a microbially produced signaling molecule, whether it is free amino acid production or other microbial metabolite, that is mitigating FGF21." I do not agree with this conclusion, because this signal cannot be there if there are no bacteria, yet the effect of PR is lost in GF mice. Instead, the data suggest that the microbiome is required for FGF21 induction (not mitigation). This then begs the question of what is happening with cellulose, with one possibility being that cellulose disrupts this normal, microbiome-dependent signal. But of course, this is all speculation because there is no insight into the underlying mechanism.

The reviewer is keen to observe that we present data whereby the complete absence of a microbiome leads to no FGF21 responsiveness to protein-restriction, and yet we also suggest that microbially derived amino acids or other metabolite can mitigate the FGF21 response when the microbiome is intact. This explanation was added to the discussion in the revision in response to the reviewer's similar previous comment: *"...Thus it seems possible that the microbiome is simultaneously necessary for the increase in FGF21 on a PR diet, and the decrease in FGF21 following cellulose. I do not think this is actually a contradiction, but I do feel the manuscript should these two observations head-on and articulate a plausible mechanism that could explain this biphasic role of the microbiome."* The reviewer had previously stated that microbially-derived amino acids was a plausible mechanism, which is indeed our central hypothesis, so in response to the above previous comment we included this explanation in our discussion, which the reviewer now disagrees with. We understand that a deeper discussion is perhaps warranted. Let us explain why we still believe microbially-derived amino acids is a plausible mediator, despite our data showing that microbiota both required for the FGF21 response and can at the same time mitigate it.

The germ-free mice are raised from birth without a microbiome. Therefore, germ-free mice have immature development in many features of their physiology, as has been reported since the creation of germ-free models (immature immune system, incomplete gut mucosal layer, etc). Nutrient sensing and metabolism is often aberrant in germ-free mice as well. For example, GF mice are resistant to diet induced obesity, and tend to have to eat more calories early in their development to maintain the same growth trajectory as their conventional counterparts. This highlights the importance of the gut microbiome for dietary energy sensing and harvest, and has been described in the literature for over a decade. However, despite this, as long as they are maintained in germ-free conditions, they remain healthy and have large litter sizes etc, so obviously, some aspects of metabolism also function completely normally. So we run germ-free experiments to better understand which pathways are affected by lack of microbes and which are not. They serve as proofs-of-concept, but do not represent the physiological norm.

Our germ-free experiments therefore demonstrate that microbiota are required for maturation of this amino acid sensing machinery and specifically the FGF21 pathway. So we are reporting for the first time in this manuscript that among the dietary sensing defects that exist in the absence of the gut microbiome, we now must add inappropriate FGF21 signaling. However, it is from the conventional mice that we must draw the majority of our physiological conclusions.

Our previous antibiotic (AB) studies, however, are interesting to re-visit in light of this discussion. We previously included antibiotic treated mouse experiments before we had access to GF mice, but replaced with the GF experiment in the revision because they tend to be preferred over antibiotics. Our antibiotic treated mice however, can be considered an intermediate between GF and conventional mice because these would be developmentally normal mice with a significantly knocked down (but not zero) gut microbiota. In these experiments, we had also included a cellulose and inulin treatment to see if these fibers had any impact on FGF21 levels in the absence of gut microbiota. In these antibiotic-treated mice, we see blunting of the FGF21 response to PR but not as complete as with the GF mice (as noted by the spread of responses in the mice), but significantly attenuated compared to the conventional mouse. This further highlights that the microbiome is a) required for normal development of the FGF21 sensing machinery, and b) when the pathway has developed normally, it is still attuned to signals coming from the gut microbiota. Furthermore, protein levels nor fibers had impact on FGF21 levels independent of the gut microbiome. As the reviewer previously noted, there is no effect to mitigate if FGF21 isn't elevated, however, it confirms that there is not a microbiome-independent role for these fibers in altering FGF21 levels.

Therefore, the most physiological state to which our study pays the most attention is the conventional mouse, and the germ-free and antibiotic experiments are proofs of concept for the importance of the gut microbiota in mediating the FGF21 response. While we cannot pinpoint the exact byproduct of the gut microbiome that mitigates FGF21, the demonstration that the gut microbiome is central to this pathway is, in our opinion, truly novel.

3. The manuscript never demonstrates that the effect of cellulose is microbiome dependent, but only assumes this relationship. As such, the effect of cellulose and the lack of effect in GF mice may be mediated by completely different mechanisms and thus not connected.

We attempted to demonstrate this in the previous revision with the antibiotic experiment as shown above. Typically for diet x microbiome studies, when a dietary treatment appears to alter a host readout, and a link to the microbiome is being made, then it is imperative to test if the test diet can elicit the observed effects independently of the microbiome. Germ-free, gnotobiotic, or antibiotic studies can

answer this. If the physiological effect of the diet is still observed in the absence or attenuation of microbiome colonization, then the dietary effect is deemed to be microbiome-independent. However, if the effect is totally lost, then we know the microbiome is a required component. The antibiotic groups above demonstrate this in our study. However, we did not include fiber groups in our germ-free experiments because our antibiotic experiment taught us that there is no FGF21 response to mitigate, which the reviewer also pointed out in the previous round of comments and suggested these groups were not useful. However, we did not know this would be the result at the start, so we had to include the fiber groups to be sure.

5. Although the manuscript does a much better job of providing clear controls, it still is missing a PS +15% Cellulose group. While not necessarily relevant for FGF21, it would be useful in other ways.

Based on the additional controls we've added to this revised manuscript, we believe that the addition of a PS+15% cellulose group would not provide further insight to our existing readouts because the PS +5% cellulose diet never created any different response compared to PS + 0% fiber. Unlike with protein deficiency, when the protein is sufficient, there isn't a context for mitigation.

6. The inulin group is particularly unusual and it is not clear how valuable this group is for comparisons. First, 5% inulin alone suppresses growth equivalently to PR, and there is no 15% inulin alone group. Second, inulin alone also increased liver FGF21, increased p-eIF2a, and increased ASNS. Thus inulin alone was sufficient to trigger a PR-like response in the liver. As such, interpretation of the PR+inulin groups is problematic, because it is difficult to know what is inulin and what is PR. The result is that inulin does not offer any value as a comparator for cellulose, and the manuscript lacks any clear explanation for why inulin is producing these divergent effects

The reviewer thoughtfully points out that inulin is inadequate at suppressing the FGF21 response to PR, especially in comparison to cellulose, which is a central point of the study that one type of fiber can rescue the PR response but the other cannot. We also feel that we can indeed distinguish what is inulin and what is PR because we have a PR + 0% fiber which is directly comparable to PR + 5 or 15% inulin. Also, we cannot create a diet made of only inulin because an animal cannot subsist on this.

From a microbial standpoint, we had to design the study with at least two structurally different types of fiber, one that is considered a microbially fermentable fiber (inulin) and one that is considered microbially non-fermentable (cellulose), but at equal concentrations on a PR diet. If we only included one fiber type, we would not have a control for it, and therefore could not determine whether the results were unique to fiber in general, or unique to that fiber type. Thus, including two very different fibers was essential for study design.

7. Interpretation of the food intake data in Fig 2d is problematic. 15% cellulose appears to completely block the PR-induction of FGF21, yet it fails to block the increase in food intake. Conversely, inulin supplementation fails to block the increase in serum FGF21, but at all doses blocks the increase in food intake. These observations thus contradict recent evidence suggesting that FGF21 drives PR-induced hyperphagia, although it is certainly possible that FGF21 is not the primary driver of PR-induced hyperphagia.

While the reviewer points out some inconsistencies between our feeding behavior data and that of the literature, we do want to point out that the 15% cellulose group in Fig 2d does in fact show no increase in food intake compared to protein sufficient groups. The asterisk signifies a significant difference

between the PR+15% cellulose and PR+15% inulin groups, not between PS +0% fiber. Similarly, the reduction of food consumption in the inulin groups is only significant within the PR groups, but not in comparison to PS. In other words, the only protein-restricted group to show a significant change in feeding behavior compared to protein sufficiency, is the PR+0% fiber group, which showed significantly increased food consumption, and this group shows the most consistent increase in plasma and gene expression of FGF21, which would be consistent with the literature. The addition of the two fibers creates more variability in the response which we acknowledge, but this has not been studied in the literature before so we are unclear whether we should expect an identical response to the existing literature. We hope this provides support for other groups, including ourselves, to further explore what factors are influencing the feeding behavior when FGF21 is altered.

8. The study in germ-free mice is nice and provocative, but also incomplete. First, the experimental only compares conventional and GF mice on PR, and thus lacks dietary controls. Ideally, the experiment should have been a 2x2 with both diet and germ-free status as effects (4 total treatments). The lack of controls is important because it is hard to define the extent to which the GF mice fail to respond to PR without a control for comparison. As such, it is not clear if the response to PR is completely lost or partially blunted. It also seems possible that GF mice could have baseline differences in important endpoints even in the absence of PR, although I acknowledge that I am less concerned for the molecular endpoints (FGF21, ASNS, etc), which would generally be low in mice on a control diet. Finally, although the failure of GF to respond to PR is an interesting observation, it must be acknowledged that the manuscript lacks any clear explanation for why this failure occurs.

We thank the reviewer for their constructive critique of our GF study and have thus incorporated data from GF mice on PS diets to demonstrate the 2x2 design as pointed out by the reviewer. We had included this PS group when we ran the original study, but did not include it in the previous revision because the PR group itself demonstrated the most relevant effect. But we acknowledge the value of including the PS group now and thank the reviewer for pointing this out. We have now also performed the liver gene expression analysis of 3pgd and Asns in order to draw more consistent comparisons with the conventional mouse datasets, and to highlight the response in the GF context. These are now shown in New Fig. 5. Overall, our interpretation of the data remains the same as before but is strengthened by these new data.

Reviewer #3 (Remarks to the Author):

All comments have been addressed appropriately.

We thank the reviewer for their positive review.

Reviewers' Comments:

Reviewer #1:

Remarks to the Author:

Thank you for responding to my final comment. I must say in addition that the response to referee 2 is important. I too had concluded that the route to impacts on FGF21 is likely via bacterially derived AA production under low-protein status. Hence, cellulose can be a substrate to maintain bacterial populations which produce AA and hence supplement host protein status under low-protein feeding. FGF21 is involved indirectly as it signals host protein status.

Reviewer #2:

Remarks to the Author:

I appreciate the extensive and thoughtful response that was provided by the authors. While I still have some concerns regarding the interpretation of the inulin data and the germ-free mouse study, I also recognize the novelty of this work and the technical difficulty in nailing down a specific mechanism that might explain the observations. The overall conclusion that dietary fiber prevents the effect of protein restriction is very interesting and will lead to a variety of future experiments focusing on the mechanism and implications of this effect.